# Inhibition of tau neuronal internalization using anti-tau single domain antibodies

Clément Danis [1,2,3] ✉, Elian Dupré [1,2], Thomas Bouillet[3], Marine Denéchaud[3], Camille Lefebvre[3], Marine Nguyen[1,2], Justine Mortelecque [1,2], François-Xavier Cantrelle [1,2], Jean-Christophe Rain [4], Xavier Hanoulle [1,2], Morvane Colin [3], Luc Buée [3] ✉ & Isabelle Landrieu [1,2] ✉

In Alzheimer's disease, tau pathology spreads across brain regions as the disease progresses. Intracellular tau can be released and taken up by nearby neurons. We evaluated single domain anti-tau antibodies, also called VHHs, as inhibitors of tau internalization. We identified three VHH inhibitors of tau uptake: A31, H3-2, and Z70$_{mut1}$. These VHHs compete with the membrane protein LRP1, a major receptor mediating neuronal uptake of tau. A31 and Z70$_{mut1}$ bind to microtubule binding domain repeats, which are involved in the interaction with LRP1. VHH H3-2 is the only VHH from our library that reduces the internalization of both monomeric tau and tau fibrils. VHH H3-2 binds a C-terminal tau epitope with high affinity. Its three-dimensional structure in complex with a tau peptide reveals a unique binding mode as a VHH-swapped dimer. These anti-tau VHHs are interesting tools to study tau prion-like propagation in tauopathies and potentially develop novel biotherapies.

In neurodegenerative disorders, there is a hypothesis suggesting that aggregation-prone proteins may have prion-like properties[1,2]. This hypothesis refers to the ability of certain misfolded neuronal proteins to spread/propagate their misfolding by seeding soluble homotypic protein to form protein aggregates, like observed in prion disorders[3–6]. This ability is associated with the inter-cellular propagation of proteins along the functional brain networks. This process can lead to the formation of pathological protein aggregates, such as tau filaments in neurofibrillary tangles (NFTs), which are commonly observed in diseases like Alzheimer's disease (AD) and other tauopathies. The prion-like propagation of misfolded tau is thought to contribute to the progression and spread of the pathology in the brain and the associated motor and cognitive deficits in tauopathies[7]. Multiple studies have shown that tau can be internalized by neuronal cells through various mechanisms, such as extracellular vesicles, through the formation of tunneling nanotubes, or directly under a free form, either soluble, phosphorylated, or aggregated[7,8]. The molecular mechanism(s) of this internalization and the nature of the internalized tau are still under debate, but this process appears to require either active endocytosis or micropinocytosis-mediated pathways[9]. Active endocytosis and micropinocytosis have been described as the main drivers of cellular uptake of tau fibrils, and this requires the involvement of heparan sulfate proteoglycans (HSPGs)[10–13]. On the other hand, active endocytosis drives the cellular uptake of monomeric tau, and the low-density lipoprotein receptor-related protein 1 (LRP1) has been proposed to be the major receptor that is responsible for its uptake[13,14].

Neuronal uptake of tau is thought to play a fundamental role in the development of tauopathies, and the strategies aimed at interfering with this uptake appear to be a valuable therapeutic option. In this context, a new type of immunological tool has emerged in the form of single-domain antibodies: nanobodies®, also called VHHs (for variable domain of the heavy chain of the heavy chain only antibodies). A VHH corresponds to the variable fragment of antibodies without a light chain, discovered in the *Camelidae* family in the 1990's[15]. The VHHs have shown their interest in research by targeting proteins for crystallization or cryogenic electron microscopy to help resolve their 3D structure[16–18], and in the medical field, as detection tools in imaging or as potential drug vectors for targeting specific tumor cell receptors[19,20].

[1]CNRS EMR9002 – BSI - Integrative Structural Biology, Lille, France. [2]Univ. Lille, Inserm, CHU Lille, Institut Pasteur de Lille, U1167 - RID-AGE - Risk Factors and Molecular Determinants of Aging-Related Diseases, Lille, France. [3]Univ. Lille, Inserm, CHU-Lille, U1172 - LilNCog - Lille Neuroscience & Cognition, Lille, France. [4]Hybrigenic Services, Paris, Evry-Courcouronnes, France. ✉e-mail: clement.danis@inserm.fr; luc.buee@inserm.fr; isabelle.landrieu@univ-lille.fr

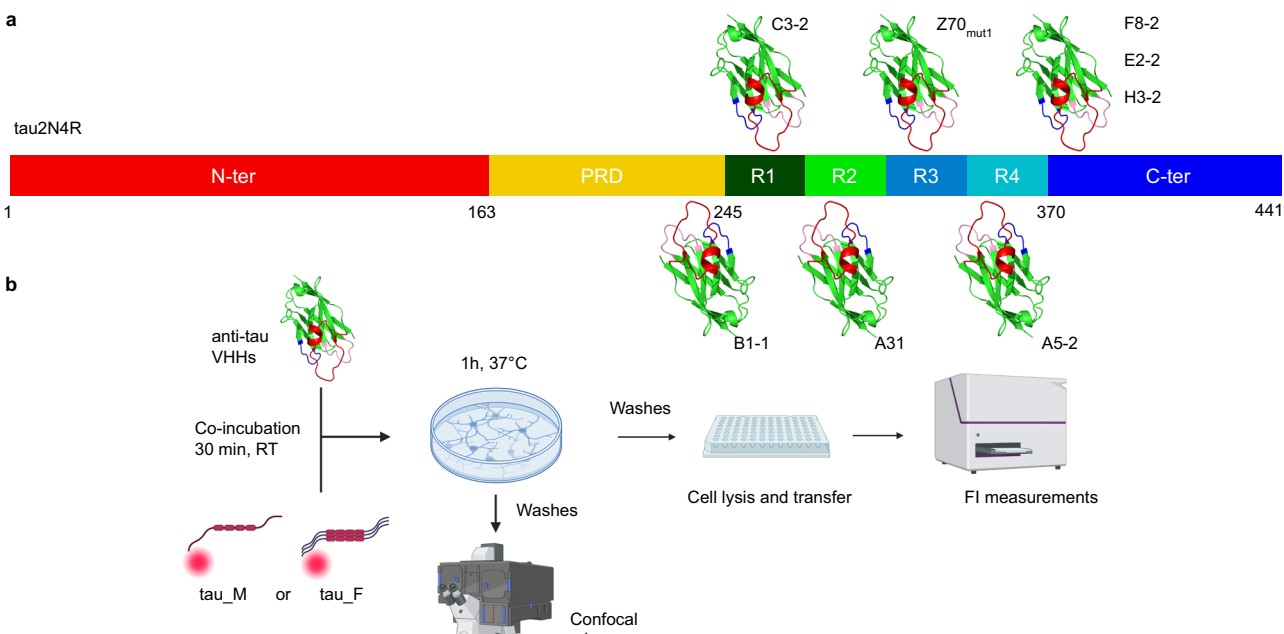

**Fig. 1 | Overview of the tau epitopes recognized by the anti-tau VHHs used in this study and schematic representation of the tau neuronal uptake assays.**
**a** Schematic representation of the tau epitopes recognized by each anti-tau VHH. VHH B1-1 binds to the tau proline-rich domain (PRD). VHH C3-2 binds to the amino-acid sequence located between the PRD and the R1 repeat of the tau microtubule-binding domain (MTBD). VHH A31 binds to the PHF6* region within the R2 repeat (Supplementary Fig. 1). VHH Z70mut1 binds to the PHF6 region within the R3 repeat. VHH A5-2 binds to the R4 repeat. VHH F8-2, E2-2, and H3-2 bind to the C-terminal domain (C-ter) of tau (Supplementary Fig. 2). The epitopes

were determined by 2D NMR interaction mapping experiments, as previously described[28–30]. **b** Schematic representation of the tau cellular uptake assay. Recombinant monomeric (tau_M) or fibrils of tau1N4R (tau_F) labeled with fluorescent dyes (Alexa546 and Atto565, respectively) and VHHs were co-incubated for 30 min at room temperature (RT). Tau-VHHs complexes were then incubated with primary neurons for 1 h at 37 °C. Finally, cells were washed and analyzed by confocal microscopy or lysed and transferred to a 96-well plate for fluorescence imaging. Created in BioRender. Tautou, M. (2025) https://BioRender.com/a27l880.

More recently, VHHs have attracted considerable interest in the study of neurodegenerative diseases such as Parkinson disease[21,22] and in the AD field with VHHs targeting the BACE1 enzyme or the Aβ amyloid peptide[23,24]. In 2016, the first specific VHH targeting the tau protein (anti-tau VHH) was generated and described, primarily with the aim of developing probes for tau detection[25]. Since this pioneering work, we and others have developed and characterized VHHs directed against different tau epitopes[26–29]. In our previous studies, we have characterized VHH F8-2 and VHH Z70, as well as their multiple variants, as powerful intracellular tools for investigating tau physiopathology and novel biotherapeutic strategies[28–30].

In this work, a series of anti-tau VHHs targeting different epitopes with a range of affinities for tau are screened, with the goal of inhibiting neuronal internalization of monomeric and fibrillar forms of tau, a critical step in the tau prion-like propagation mechanism. From this screen, the VHHs A31, Z70mut1, and H3-2 are identified as the most potent inhibitors of cellular internalization of monomeric tau in primary mouse neuronal cultures. We also confirm the inhibitory effect of VHHs A31 and Z70mut1 on the cellular uptake of the isolated microtubule-binding domain of tau (tauMTBD). The VHHs A31 and Z70mut1 can compete with the binding between tau and a LRP1 membrane receptor domain, supporting the importance of the tauMTBD for binding to LRP1 and in the internalization process of monomeric tau. Interestingly, H3-2, a C-terminal tau-binding VHH, is the best inhibitor of tau uptake in both monomeric and fibrillar forms, and the strongest competitor for LRP1 binding. Using a combination of biophysical methods, we show that VHH H3-2 forms a swapped dimer through the formation of an intermolecular β-sheet between two subunits, each binding to its C-terminal tau epitope. This binding mode may explain its potent activity in inhibiting tau cellular uptake. These VHHs thus represent alternative immunological tools that can

be used to study prion-like propagation in tauopathies, taking advantage of the ability to screen for well-defined epitopes and affinities within the VHH series.

## Results

### Biochemical characterization of VHHs that bind different regions of tau

We here used 8 VHHs that bind different regions of tau (anti-tau VHHs), including the proline-rich domain (PRD), the microtubule-binding domain (tauMTBD), and the C-terminal domain of tau (Fig. 1a). The tau binding sites of some of these VHHs have been identified and described in our previous studies: VHHs B1-1, C3-2 and A5-2 bind specifically to the PRD, the R1, and R4 repeats, respectively[29]; VHH Z70mut1 recognizes the PHF6 peptide sequence located in the R3 repeat[30] and VHHs F8-2 and H3-2 bind to the C-terminal domain of tau[28,31]. In the present study, we have also characterized the binding of two additional anti-tau VHHs, namely VHHs A31 and E2-2, to the largest tau isoform, tau2N4R. Comparison of the 2D NMR spectra of [15]N-tau2N4R alone in solution or in the presence of the VHH identified the spectral perturbations that were used to define the binding region. VHH A31 is specific for a sequence located in the R2 repeat encompassing the PHF6* peptide (Supplementary Fig. 1). VHH E2-2 binds the same C-terminal epitope as VHH F8-2, as might be expected since they share the same complementarity determining region 3 (CDR3) sequence (Supplementary Figs. 2, 3)[28,32].

We next determined the affinity of the interaction between tau2N4R, tau1N4R, and these 8 VHHs produced without any tag (Table 1 and Supplementary Figs. 4, 5). The kinetic parameters of the tau-VHH interactions were characterized using surface plasmon resonance (SPR), with biotinylated tau2N4R or tau1N4R immobilized on the surface of a streptavidin-functionalized chip. VHHs were tested as

**Table 1 | Tau epitopes (identified by NMR) recognized by the anti-tau VHHs and $k_{on}$, $k_{off}$, and $K_D$ (affinity parameters) measured for each VHH and full-length tau2N4R or tau1N4R**

| VHH | tau epitope | $k_{on}$ ($M^{-1}.s^{-1}$)*$10^{+2}$ | | $k_{off}$ ($s^{-1}$)*$10^{-3}$ | | $K_D$ (nM) | |
|---|---|---|---|---|---|---|---|
| | | 2N4R | 1N4R | 2N4R | 1N4R | 2N4R | 1N4R |
| B1-1 | $_{224}$KKVAVVR$_{230}$ | 379 ± 25 | 608 ± 18 | 10.9 ± 0.66 | 27.9 ± 0.40 | 289 ± 26 | 460 ± 15 |
| C3-2 | $_{243}$LQTAPVPMPDLKNVKSKI$_{260}$ | 29 ± 0.2 | 29 ± 0.5 | 2.8 ± 0.02 | 2.5 ± 0.04 | 979 ± 12 | 857 ± 20 |
| A31 | $_{275}$VQIINKKLDLSN$_{286}$ | 3524 ± 26 | 3311 ± 24 | 12.3 ± 0.06 | 12.1 ± 0.06 | 35 ± 0.1 | 37 ± 0.1 |
| Z70$_{mut1}$ | $_{305}$SVQIVYKP$_{312}$ | 1517 ± 30 | 1523 ± 30 | 65.8 ± 0.62 | 71.4 ± 0.65 | 434 ± 9 | 468 ± 10 |
| A5-2 | $_{339}$VKSEKLDFKDRVQSKIGSLDNITHVPGGGNKK$_{370}$ | 977 ± 41 | 618 ± 25 | 56.2 ± 1.15 | 32.4 ± 0.62 | 576 ± 27 | 525 ± 24 |
| F8-2 | | 531 ± 11 | 575 ± 11 | 40 ± 0.40 | 42.5 ± 0.39 | 753 ± 17 | 740 ± 16 |
| E2-2 | $_{369}$KIETHKLTFREN$_{381}$ | 449 ± 3 | 467 ± 3 | 3.8 ± 0.03 | 4.5 ± 0.04 | 84 ± 1 | 96 ± 1 |
| H3-2 | | 6298 ± 31 | 5544 ± 25 | 5.6 ± 0.03 | 6.0 ± 0.03 | 8.9 ± 0.1 | 11 ± 0.1 |

analytes in single-cycle kinetics (SCK), which consists of increasing the analyte concentration in five consecutive injections, with a short dissociation time and no regeneration between injections. No significant differences were observed in the affinity parameters of these VHHs between the two tau isoforms tested, which was expected since all of these anti-tau VHHs bind outside the tau N-terminal domain. The $K_D$s of the VHH interactions ranged from ~ 10 to 750 nM for tau, with VHHs H3-2, A31 and E2-2 showing the best affinity with $K_D$s below the 100 nM range (Table 1). Interestingly, VHH E2-2 had an 8-fold higher affinity for tau than VHH F8-2 (Table 1 and Supplementary Figs. 4, 5), despite sharing the same CDR3 recognition loop[28]. The $k_{off}$, which is an interesting kinetic parameter as it relates to the residence time of the VHH on the tau protein, showed the best values for the VHHs C3-2, E2-2, and H3-2 ($k_{off}$ tau1N4R = $2.5*10^{-3}$, $4.5*10^{-3}$ and $6*10^{-3} s^{-1}$, respectively, Table 1).

### VHHs H3-2, A31 and Z70$_{mut1}$ inhibit monomeric tau1N4R cellular uptake

Next, we evaluated the potential of each anti-tau VHH to interfere with the neuronal internalization of free tau in mouse primary cultured neurons. To this end, we designed a cellular tau uptake assay based on the quantification and visualization of the intracellular fluorescence of Alexa546-labeled tau after incubation with primary neuronal cultures in the presence of each anti-tau VHH or different controls (Fig. 1b). We used as negative control an irrelevant green fluorescent protein (GFP)-targeting VHH. As positive controls, we selected two described inhibitors of tau cellular uptake, that are the LDL receptor-associated protein (RAP) and heparin. RAP protein is a partner of the LRP1 receptor and can compete with tau-LRP1 binding[14,33] while heparin can compete with tau-HSPGs at the membrane surface[10–13]. In the presence of tau1N4R pre-incubated in PBS only (tau_M), internalization, measured as a fluorescence readout after cell lysis, is normalized to 100% uptake (Fig. 2a). Under this condition, fluorescence imaging of neurons revealed intracellular tau signal observed as numerous red puncta throughout the cells (Fig. 2b). RAP, but not heparin, significantly reduced tau cellular uptake (31 ± 14% and 88 ± 23%, respectively) while VHH anti-GFP did not interfere (Fig. 2a). At 30, 100 and 300 nM of VHH, we observed a significant dose-dependent reduction in tau cellular uptake in the presence of the VHHs A31 (80 ± 19%, 60 ± 18% and 47 ± 24%, respectively) and H3-2 (53 ± 16%, 38 ± 8% and 23 ± 8%, respectively), the latter one showing an efficacy similar to that of RAP (Fig. 2a and Supplementary Fig. 6a, b). It was only at 300 nM VHH that the VHH Z70mut1 also significantly blocked the cellular uptake of tau, while the other VHH tested remained inactive (Supplementary Fig. 6a). Consistent with the measurements, visualization of the neuronal cells showed a reduction in the intracellular tau signal when labeled tau1N4R was co-incubated with RAP, VHH A31 or H3-2, and to a lesser extent VHH Z70$_{mut1}$ (Fig. 2b and Supplementary Fig. 7).

In summary, we have identified VHHs A31, H3-2, and, to a lesser extent Z70$_{mut1}$, from our anti-tau VHH collection as the most potent inhibitors of cellular tau internalization.

### VHHs A31 and Z70$_{mut1}$ bind monomeric tauMTBD and inhibit its cellular uptake

In addition, we performed cellular uptake assays using only the recombinant tauMTBD, which can be internalized into cells as the recombinant full-length monomeric tau[14]. The interaction of VHHs with tauMTBD was first characterized by SPR using biotinylated tauMTBD immobilized on the surface of a streptavidin-functionalized chip (Supplementary Fig. 8). SCK experiments confirmed the binding of VHHs A31 and Z70$_{mut1}$ to tauMTBD, with $K_D$s of 41 and 799 nM, consistent with their respective binding affinities for tau1N4R. As expected, we did not detect any interaction between tauMTBD and the two VHHs E2-2 and H3-2, consistent with their identified epitope (Fig. 1a and Supplementary Fig. 8). We then performed the cellular uptake assay based on quantification of the total intracellular fluorescence signal after cell lysis, using labeled Atto488-tauMTBD pre-incubated with each of these four VHHs or under control conditions (Fig. 3a). The measurements showed a significant inhibition of the uptake by VHHs A31 and Z70$_{mut1}$ (36 ± 11% and 61 ± 16%, respectively), whereas VHHs E2-2 and H3-2 had no significant effect (103 ± 14% and 94 ± 13%, respectively; Fig. 3a). Accordingly, fluorescence imaging of neurons revealed intracellular tauMTBD signal observed as numerous green puncta throughout the cells under control conditions (anti-GFP) and in the presence of VHHs E2-2 and H3-2 (Fig. 3b). Conversely, in cells co-incubated with tauMTBD in the presence of VHHs A31 or Z70$_{mut1}$, a reduction in intracellular tauMTBD signal is observed, with the strongest effect detected with VHH A31 (Fig. 3b).

These data demonstrate that VHHs A31 and Z70$_{mut1}$ block cellular internalization of tauMTBD, but not VHHs E2-2 and H3-2, strongly suggesting that the specific binding of the VHHs to their tau epitope is required to achieve the uptake inhibition.

### VHHs H3-2 inhibits fibrillar tau1N4R cellular uptake in mouse primary cultured neurons

We then repeated the uptake experiments using fibrillar recombinant tau1N4R, labeled at the lysine residues with the fluorescent dye Atto565 prior to aggregation into fibrils (Supplementary Fig. 9). 200 nM of these labeled tau1N4R fibrils was co-incubated with either PBS (tau_F), the controls (RAP, heparin, VHH anti-GFP), or the anti-tau VHHs (Fig. 1b, Fig. 4a) prior to dilution into the medium of primary neuronal cultures. In the presence of tau1N4R fibrils co-incubated with PBS only (tau_F), internalization of labeled tau was measured and normalized to 100% uptake (100 ± 12%, Fig. 4a). Interestingly, in contrast to tau monomers, heparin but not RAP significantly reduced tau cellular uptake (38 ± 12% and 92 ± 21%, respectively), whereas the anti-GFP VHH had no effect (96 ± 27%; Fig. 4a). From our collection of

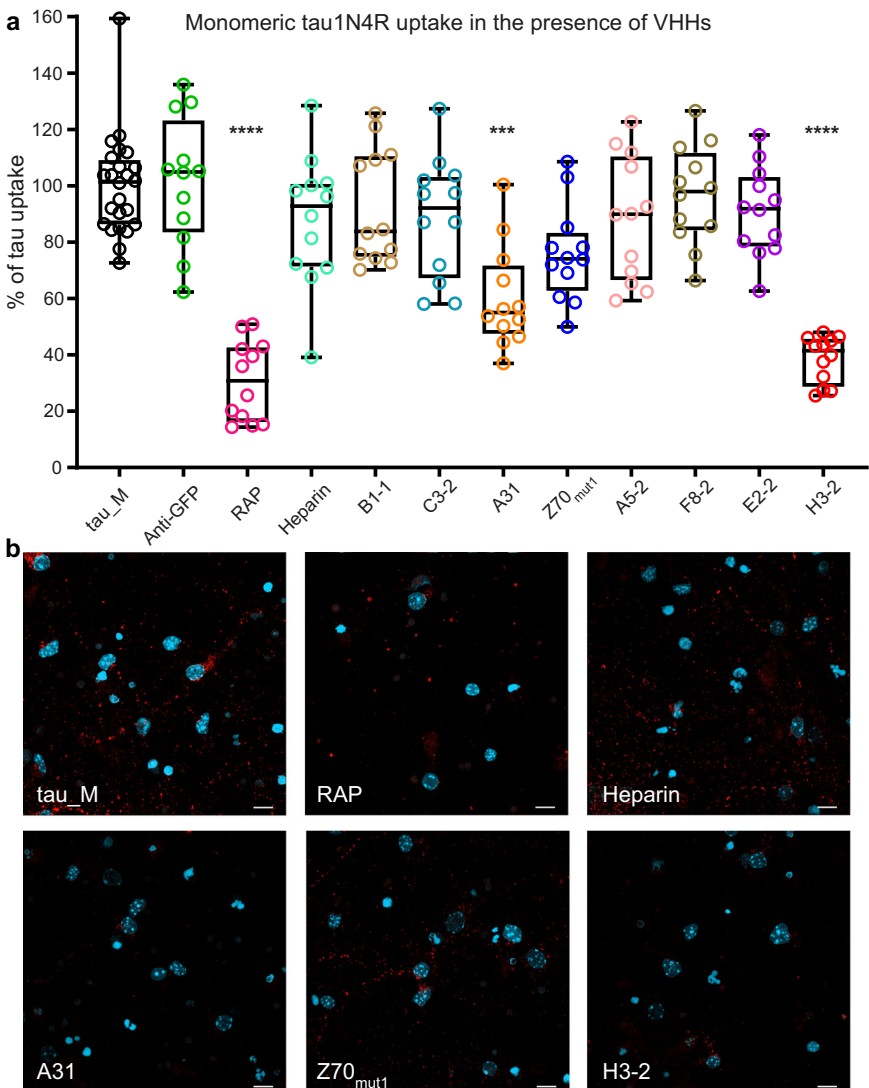

**Fig. 2 | Cellular internalization of monomeric tau is inhibited by anti-tau VHHs A31 and H3-2. a** Inhibition of cellular uptake of monomeric tau1N4R by the anti-tau VHHs. Percentage of cellular tau uptake based on Alexa546 fluorescence signal coming from monomeric tau1N4R (50 nM) co-incubated with PBS (tau_M), RAP (100 nM), heparin (50 μg/mL), and the VHHs (100 nM). $n = 4$ independent experiments per condition. Data were normalized to the condition of tau1N4R pre-incubated with PBS as 100% of tau uptake. Box plots show the median (center line), 25th, 75th percentiles (box), and minimum and maximum values (whiskers), all data points are shown (single points). Data were analyzed using one-way nonparametric ANOVA (Kruskal-Wallis) with Dunn's multiple comparison test ****$p < 0.0001$; ***$p < 0.001$). RAP, VHHs A31 ($p = 0.0004$), and H3-2 significantly reduced cellular tau uptake. **b** Confocal microscopy analysis of cellular tau uptake assay using recombinant tau1N4R labeled with Alexa546 dye and in the presence of PBS (tau_M), RAP, heparin, and VHHs A31, Z70mut1 or H3-2 (images are representative of a $n = 2$ independent experiments per condition). Tau was visualized in red, and nuclei in light blue. The scale bar is 10 μm.

anti-tau VHHs, only VHH H3-2 significantly reduced the cellular uptake of tau1N4R fibrils (68 ± 20%; Fig. 4a). Accordingly, intracellular tau was observed as a strong red carpet fluorescence signal throughout the cells under control conditions (tau_F, anti-GFP), in the presence of RAP and all VHHs tested except H3-2 (Fig. 4b and Supplementary Fig. 10). In contrast, primary neuronal cell cultures co-incubated with heparin showed a strong reduction in intracellular tau fibril signals, whereas VHH H3-2 caused a moderate reduction, consistent with the results obtained by measuring the fluorescence in the cell extract (Fig. 4b). We also verified the binding of VHH H3-2 to tau fibrils by coupling VHH H3-2 to nanogold particles for visualization by transmission electron microscopy. The functionalized particles were detected along the entire length of the tau fibers, validating the binding of VHH H3-2 to the recombinant fibrils used in the cellular uptake assay (Supplementary Fig. 11). Based on all these measurements, we identified VHH H3-2 as a binder to and as an inhibitor of cellular internalization of recombinant tau fibrils.

## VHHs A31, Z70mut1 and H3-2 compete with the binding between tau1N4R and cluster III domain of the neuronal membrane receptor LRP1

Since the inhibition of cellular tau uptake seemed related to the direct binding of the VHHs, we reasoned that competition with receptor interaction(s) at the membrane may be involved. The uptake of recombinant tau and tauMTBD fragments has been described to be mainly mediated by the membrane receptor LRP1[14]. In addition, tau-LRP1 in vitro interaction was characterized using SPR and showed that tau binds to LRP1 motifs called clusters II, III, and IV with the same affinity[14,33]. Anti-tau VHHs mechanism of action was thus investigated by performing a competition assay between one of the LRP1 clusters (cluster III), tau, and the VHHs (Fig. 5). We first validated the binding of recombinant tau1N4R to LRP1 cluster III ($K_D$ of 91 ± 1 nM, Supplementary Fig. 12a). We also showed that tau binding to LRP1 cluster III was abolished in the presence of human RAP protein (Supplementary Fig. 12b). Finally, we performed the competition assay by comparing the response curve of 100 nM of

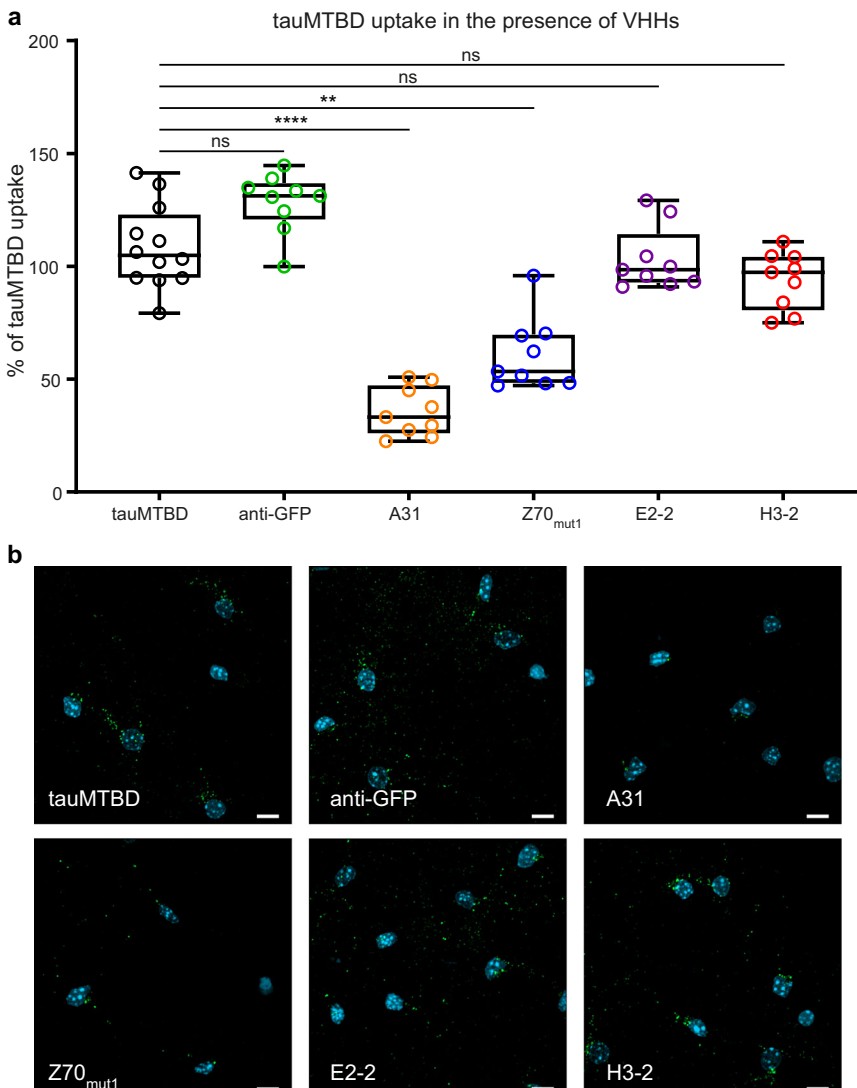

**Fig. 3 | VHHs A31 and Z70mut1 block the cellular uptake of recombinant tau microtubule binding domain (tauMTBD).** Cellular tauMTBD uptake assay performed using 0.5 μM of recombinant monomeric tauMTBD labeled with the Atto488 fluorophore. **a** Percentage of cellular tauMTBD uptake based on Atto488 fluorescence signal quantification in the conditions resulting from tauMTBD co-incubation with each of the VHHs anti-GFP, A31, Z70mut1, E2-2, and H3-2 at a concentration of 1 μM. $n = 3$ independent experiments per condition. Data were normalized to the percentage of uptake relative to the condition of tauMTBD pre-incubated with PBS, defined as 100% of tauMTBD uptake. Box plots show the median (center line), 25th, 75th percentiles (box), and minimum and maximum values (whiskers), all data points are shown (single points). Data were analyzed using one-way nonparametric ANOVA (Kruskal-Wallis) with Dunn's multiple comparison test (****$p < 0.0001$; **$p < 0.01$). VHHs A31 and Z70mut1 ($p = 0.0061$) significantly reduced cellular tau uptake. **b** Confocal microscopy analysis of cellular tauMTBD uptake assay under the conditions described in (**a**) (images are representative of a $n = 2$ independent experiments per condition). TauMTBD is visualized in green, and nuclei in light blue. The scale bar is 10 μm.

recombinant tau1N4R binding to immobilized LRP1 cluster III in the absence and presence of increasing concentrations of VHH A31, Z70mut1, E2-2 and H3-2 (Fig. 5a–d). VHH E2-2 did not affect the binding response of tau to LRP1 cluster III (Fig. 5a). Interestingly, VHHs Z70mut1 and A31 both reduced the binding response of tau to LRP1 cluster III in a concentration-dependent manner, demonstrating their ability to compete with tau binding to LRP1 cluster III (Fig. 5b, c). As observed in the cellular uptake assays, VHH A31 was a better inhibitor of tau binding than VHH Z70mut1. Finally, VHH H3-2, the strongest tau uptake inhibitor, also caused the strongest competition effect (Fig. 5d). Taken together, the data suggested that VHHs A31, Z70mut1, and H3-2 compete with tau binding to the LRP1 receptor, thereby blocking monomeric tau cellular uptake.

## Comparison of the interaction of VHHs E2-2 and H3-2 with their tau epitope

Consistent with our findings, it has previously been proposed that LRP1 interaction with tau is mediated by the tauMTBD[14,33]. However, the

C-terminal domain of tau, recognized by both VHHs E2-2 and H3-2, is not known to be involved in tau recognition by LRP1. To clarify this point, we next evaluated the mechanism of action of VHH H3-2 by comparing its strong inhibitory activity with that of VHH E2-2, which conversely did not show any significant activity against tau uptake and did not compete with LRP1 (Figs. 2 and 5). Although their differences in affinity for tau might play a role (Table 1), the striking inhibition differences were likely not limited to this. We, therefore, investigated the tau binding site(s) of these two VHHs. VHH H3-2, epitope mapping by NMR was not as precise as for VHH E2-2 (Supplementary Fig. 13). Indeed, the NMR signal broadening upon the addition of VHH H3-2 corresponded to larger regions of the tau protein that include the microtubule-binding domain. Nevertheless, we still identified the same C-terminal region with the major broadened resonances corresponding to the tau[369–381] peptide, 369KKIETHKLTFREN381, which includes the minimal 373THKLTF378 epitope (Supplementary Fig. 13)[31]. Based on these interaction data, we examined the binding of the two VHHs to the tau-C-ter

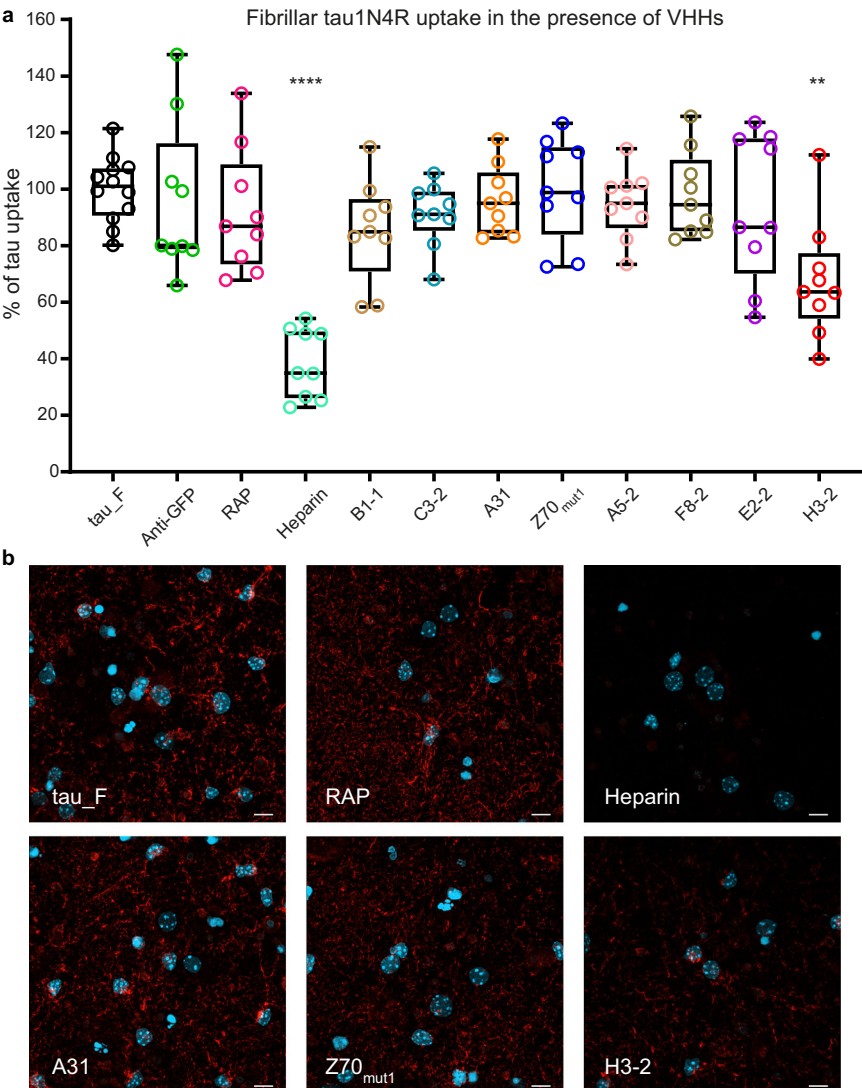

**Fig. 4 | Cellular internalization of tau fibrils is inhibited by the anti-tau VHH H3-2. a** Inhibition of cellular uptake of fibrillar tau1N4R by the anti-tau VHHs. Percentage of cellular tau fibril uptake based on Atto565 fluorescence signal coming from tau1N4R fibrils co-incubated with PBS (tau_F), RAP (400 nM), heparin (50 μg/mL), and the VHHs (400 nM). $n = 3$ independent experiments per condition. Data were normalized to the condition of tau1N4R fibrils pre-incubated with PBS as 100% of tau uptake. Box plots show median (center line), 25th, 75th percentiles (box,) and minimum and maximum values (whiskers), all data points are shown (single points). Data were analyzed using one-way nonparametric ANOVA (Kruskal-Wallis) with Dunn's multiple comparison test (****$p < 0.0001$; **$p < 0.01$). Heparin and VHH H3-2 ($p = 0.0096$) significantly reduced cellular tau fibril uptake. **b** Confocal microscopy analysis of fibrillar tau cellular assays using recombinant tau1N4R fibrils labeled with Atto565 dye and in the presence of PBS (tau_M), RAP, heparin, and VHHs A31, $Z70_{mut1}$, or H3-2 (images are representative of a $n = 2$ independent experiments per condition). Tau was visualized in red, and nuclei in light blue. The scale bar is 10 μm.

peptide using biotinylated VHH immobilized on the surface of a streptavidin-functionalized chip. The tau-C-ter peptide was used as the analyte in multi-cycle kinetics (MCK) consisting of nine or ten different cycles, each cycle corresponding to an injection of increasing peptide concentration (Fig. 6a). MCK experiments provide all the kinetic parameters, but in addition, $K_D$s were alternatively determined using steady-state analysis by reporting the maximum response of binding for each peptide concentration (Table 2). The stoichiometric ratio (SR), which provides an estimate of the stoichiometry of the interaction, was also calculated and presented similar values (Fig. 6b and Table 2). In this experimental setup, VHH E2-2 showed a 1.7 to 2-fold higher $K_D$ for the peptide than VHH H3-2, ($KD_{E2-2} = 518–820$ nM and $KD_{H3-2} = 297–380$ nM; Fig. 6a, b and Table 2). Thus, we confirmed the NMR epitope mapping by showing that both VHHs bind to the tau-C-ter peptide, with the highest affinity determined with H3-2. However, those differences in affinities do not provide a compelling basis to explain the observed differences in their respective inhibitory activities.

## VHH H3-2 bound to tau[369–381] C-terminal peptide forms a dimer in crystals and in solution

Since neither affinity nor epitope alone could explain by themselves such a strong difference in the inhibition of cellular tau uptake by VHHs E2-2 and H3-2, we next turned to conformational aspects of the interaction between tau and these two VHHs. We performed co-crystallization assays of the VHHs H3-2 and E2-2 in the presence of the tau-C-ter peptide. We solved the structure of VHH H3-2 bound to the tau-Cter peptide by X-ray crystallography, at a resolution of 1.8 Å (PDB 9G13), while crystallization assays of the VHH E2-2 complex proved unsuccessful (Supplementary Table 1). Surprisingly, the structure of this complex revealed a swapped dimer of two molecules of VHH H3-2, formed by the interaction of their CDR3 in an anti-parallel β-sheet conformation and the exchange of their last β-strand (corresponding to framework 4 of the VHH) (Fig. 6c). The interaction with the tau-Cter peptide was almost restricted to the CDR3, with one peptide interacting with one VHH. The interaction was mainly driven by main-chain

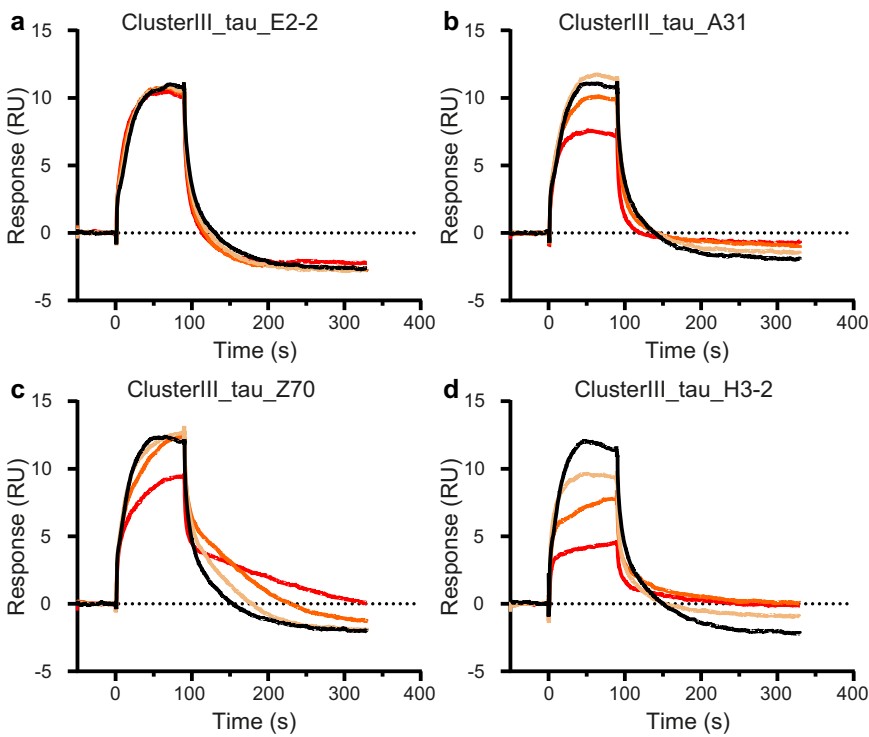

**Fig. 5 | VHHs A31, Z70$_{mut1}$ and H3-2 compete for the interaction between monomeric tau1N4R and LRP1 cluster III in vitro.** Sensorgram (control subtracted data) of tau1N4R binding to immobilized LRP1 cluster III in the absence (black curve) or the presence of 30 (yellow curve), 100 (orange curve), and 300 nM (red curve) of (**a**) VHH E2-2, of (**b**) VHH A31, of (**c**) VHH Z70$_{mut1}$ and of (**d**) VHH H3-2.

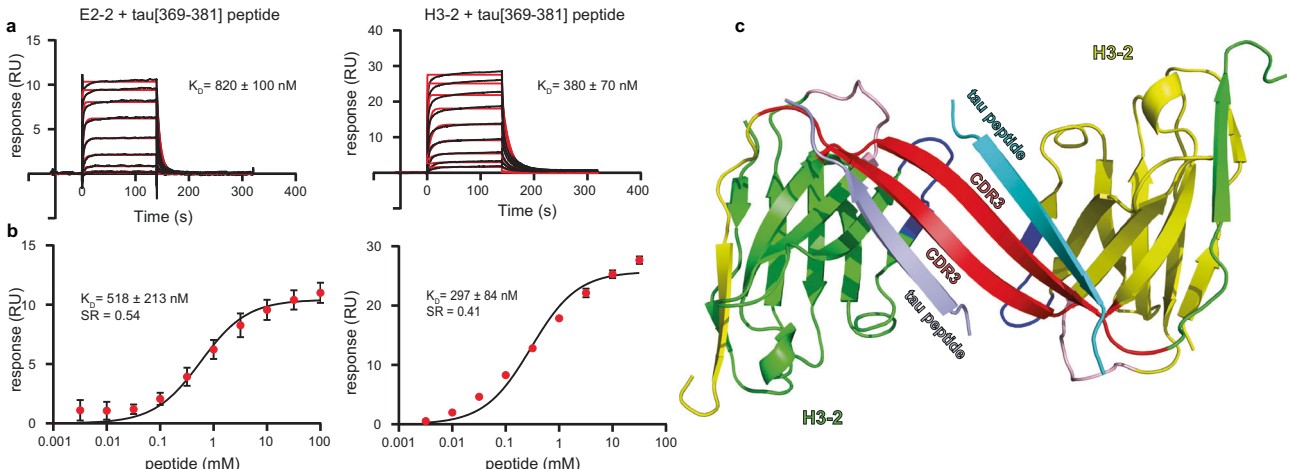

**Fig. 6 | VHH H3-2 binds with high affinity to its tau C-terminal epitope and dimerizes in vitro. a** Sensorgrams (reference subtracted data) of multi-cycle kinetics (MCK) analysis performed on immobilized biotinylated VHH E2-2 and H3-2, with injections of increasing concentrations of tau[369−381] C-terminal peptide (from 3 nM to 100 μM and from 3 nM to 32 μM, respectively; $n = 3$ independent experiments). Black lines correspond to the fitted curves, red lines correspond to the measurements. K$_D$s were obtained using the kinetic model and are presented as mean values ± standard deviation (**b**). The maximum response (RU) observed for each peptide concentration was plotted and represented as a concentration-response curve (CRC). Black lines correspond to the fitted curves, red dots to the mean of the maximum response (in RU), from experiment (**a**) for each peptide concentration. Error bars represent standard deviation. The K$_D$ and stoichiometric ratio (SR) values extracted from these data are shown. K$_D$s were obtained using the steady-state fitting model and are presented as mean values ± standard deviation. The SR have been calculated from the Rmax of the biological replicate represented here. **c** Ribbon representation of the crystal structure of the complex between VHH H3-2 and the tau[369−381] C-terminal peptide. The CDR1, CDR2, and CDR3 loops of VHH H3-2 are colored in pink, in dark blue, and in red, respectively. The framework regions of the VHHs are shown in green for one subunit and yellow for the other. The tau[369−381] peptides are shown in cyan and light blue.

hydrogen bonds between the peptide and the CDR3 in a parallel β-sheet conformation, forming a 4-stranded β-sheet with 2 CDR3 and 2 peptides. The core of the interaction site was between residues 105–113 of the CDR3 and residues 371-379 of the tau-C-ter peptide (Supplementary Fig. 14). While the interaction was well defined in the structure, some loop regions, including the CDR1 and 2 loops, (residues 26–35, 54–60, 76–81, 114–121) were poorly resolved and contained Ramachandran outliers.

To verify that this dimer was not due to crystallographic constraints, we used size exclusion chromatography (SEC), in the absence or presence of the tau-C-ter peptide at a 1: 2 ratio (20 μM final concentration of VHH; 40 μM final concentration of tau-Cter peptide) (Fig. 7a and

**Table 2 | Table summarizing the affinity parameters of the interaction between the tau[369–381] C-terminal peptide and VHHs E2-2 or H3-2**

| VHH | Kinetic analysis | | | Steady-state analysis | |
|---|---|---|---|---|---|
| | $k_{on}$ (M$^{-1}$.s$^{-1}$)*10$^{+2}$ | $k_{off}$ (s$^{-1}$) *10$^{-3}$ | $K_D$ (nM) | $K_D$ (nM) | SR |
| E2-2 | 1540 ± 271 | 124 ± 13 | 820 ± 100 | 578 ± 213 | 0.45 |
| H3-2 | 1918 ± 359 | 71 ± 2 | 280 ± 70 | 297 ± 84 | 0.41 |

Kinetic $k_{on}$ and $k_{off}$, and $K_D$ values are included in the table. For the steady-state analysis, $K_D$ were determined using the one-site specific binding model. The stoichiometric ratio (SR) was calculated from the ratio of the experimental Rmax to the theoretical Rmax.

Supplementary Fig. 15). In the absence of the tau-C-ter peptide, we identified a major peak of the VHH E2-2 with an elution volume corresponding to an estimated molecular weight (MW) of 17.2 kDa (blue curve), compatible with a monomer (theoretical MW of 15 kDa). Interestingly, injection of VHH H3-2 without peptide yielded a major peak with an elution volume corresponding to a MW of 34 kDa (red curve), compatible with the size of a dimer (theoretical MW of 29 kDa). An additional peak with an elution volume corresponding to a MW of 14.1 kDa (red curve), compatible with the size of a monomer (theoretical MW of 14.5 kDa), was also observed. In the presence of tau-Cter peptide, the elution profiles of VHH E2-2 and H3-2 were both conserved (green and yellow curves, respectively). These data suggest that while VHH E2-2 may predominantly adopt a monomeric form in solution, VHH H3-2 may spontaneously adopt a dimeric form independent of binding to tau. In addition, we designed and performed a homogeneous time-resolved fluorescence (HTRF) assay using recombinant VHHs and tau-C-ter peptides. Unlike SEC analysis, the HTRF assay requires low protein concentrations (low nM range) to study protein-protein interactions. Using VHHs with a 6-histidine tag in the N-terminal position, 6His-VHH H3-2 or 6His-VHH E2-2 were then bound by an anti-6His antibody coupled to either the fluorescence donor (Terbium) or the fluorescence acceptor (D2). We then co-incubated both fluorescence acceptor-coupled VHH H3-2 or VHH E2-2 with their donor counterparts, and we performed a concentration-response experiment using increasing concentrations of the tau-C-ter peptide (Fig. 7b). Addition of tau-Cter peptide to the VHH E2-2 did not induce an increase in the HTRF ratio signal (Fig. 7c), in contrast to the case of VHH H3-2. We concluded that the increase in the HTRF ratio signal was related to the peptide binding, which induced dimerization of VHH H3-2 with an EC50 of 2.7 ± 0.9 μM (Fig. 7c). Taken together, these experiments strongly support that VHH H3-2 dimerization is concentration dependent and that its binding to the C-terminal domain of tau may lower the dimerization concentration.

## Discussion

The prion-like propagation hypothesis has been proposed to explain the spread of tau pathology in the brain observed in Alzheimer's disease. The endocytic pathway has been identified as key to enabling tau entry into neurons, although the cellular uptake of the different tau species is not fully understood[34–36]. Receptor-mediated entry has been proposed for monomeric tau, involving its interaction with the neuronal membrane receptor low-density lipoprotein receptor-related protein 1 (LRP1) and the sortilin-related receptor 1 (SORL1, SORLA, or LR11), a protein structurally related to LRP1 and also known to be genetically associated to AD[14,37]. Heparan sulfate proteoglycans also play an important role, especially regarding the uptake of recombinant tau fibrils, and may act as co-receptors in concert with these membrane receptors[10,11,33,38]. However, the details of molecular recognition remain elusive. Thus, the mechanism of the cellular uptake of tau and the subsequent development of related therapeutic modalities remain major questions in the field[9].

Based on the tau prion-like propagation hypothesis, binding to and clearance of extracellular tau protein may be key for preventing neuronal internalization and slowing disease progression. The free forms of extracellular tau protein account for 90% of the total tau found in the extracellular space[39,40]. Several anti-tau immunotherapies targeting different tau epitopes are currently in human clinical trials[41]. The mode of action of these antibodies is still not fully understood, but one hypothesis is that tau-antibody complexes may form in the extracellular space, blocking tau entry and allowing its subsequent clearance from the brain. Recent work has shown that tau antibodies targeting the extracellular tau species may also be internalized together with tau. These complexes would be taken up by the cytosolic receptor TRIM21, resulting in the neutralization of the seeding capacity through the proteasome targeting[42–44]. However, some small tau fibrils can still escape the TRIM21 pathway and nucleate soluble tau to seed further aggregation[45]. It is also unknown which internalized tau species contribute the most to intracellular tau seeding. Many different forms of soluble tau, not just oligomeric tau species, have been proposed to induce intracellular tau seeding[38,46–48].

Here, we took advantage of anti-tau VHHs targeting different tau epitopes to further investigate their ability to block tau entry into mouse primary cultured neurons. We first confirmed the effect of two inhibitors of tau neuronal uptake, the RAP protein and heparin. Interestingly, RAP and heparin, well-described competitors of either the LRP1 receptor or the HSPGs, are here positive inhibitors of cellular uptake of either monomeric tau or tau fibrils, respectively. These data are consistent with a recent study using pluripotent stem cell-derived neurons (iPSCNs), which confirmed these different mechanisms for the neuronal uptake of recombinant tau and tau fibrils[13]. We then performed direct uptake of recombinant monomeric or tau fibrils, co-incubated with anti-tau VHH, and measured the resulting tau uptake. We have observed striking differences in the ability of the VHHs to block cellular uptake of monomeric tau, with the most potent ones corresponding to VHHs H3-2, A31, and, to a lesser extent Z70$_{mut1}$.

VHHs A31 and Z70$_{mut1}$ both target the microtubule-binding domain of tau (tauMTBD), more specifically, the PHF6* and PHF6 peptides, respectively (Supplementary Fig. 1)[30]. The membrane receptor LRP1 has been shown to interact directly, and with similar affinities, with both full-length tau and the tauMTBD fragment, suggesting that the tauMTBD is sufficient to mediate tau binding to LRP1[14,33]. We confirmed these results using the cluster III domain of LRP1, with the same range of reported affinities under the conditions of our assay (Supplementary Fig. 12). The in vitro competition of VHHs A31 and Z70$_{mut1}$ with tau for the interaction with LRP1 cluster III (Fig. 5) suggested that they could act as tau-LRP1 competitors at the cell membrane and prevent tau receptor-mediated endocytosis. The strongest effect was observed with VHH A31, which could be explained by the 10-fold strongest affinity for tau1N4R compared to VHH Z70$_{mut1}$ (Table 1).

Unexpectedly, VHH H3-2, which targets a C-terminal tau epitope outside the tauMTBD, was found to be the most potent of all VHHs in blocking monomeric tau entry. In addition, VHH H3-2 is the only VHH in our collection that also shows a moderate but significant reduction in tau fibril uptake. In parallel, VHHs F8-2 and E2-2, which also target the tau C-terminal domain, do not block tau cellular uptake. VHH H3-2 shares the same epitope, although its sequence is completely different (Supplementary Fig. 3), but its affinity for tau is 9 and 67-fold higher than that of VHHs E2-2 and F8-2, respectively (Table 1). These data, therefore, suggest that affinity is an important contributor to the activity of the VHHs in blocking cellular uptake of tau, but other factors may also contribute. Indeed, VHH H3-2 showed the strongest competitive effect on tau-LRP1 binding in vitro in SPR assays, despite the fact that the C-terminal domain of tau is not described to interact with LRP1, whereas VHH E2-2 remained inefficient. This prompted us to further investigate the binding mode of VHH H3-2 to tau. The three-dimensional X-ray structure of VHH H3-2 and biophysical assays showed that it can spontaneously form dimers in solution and

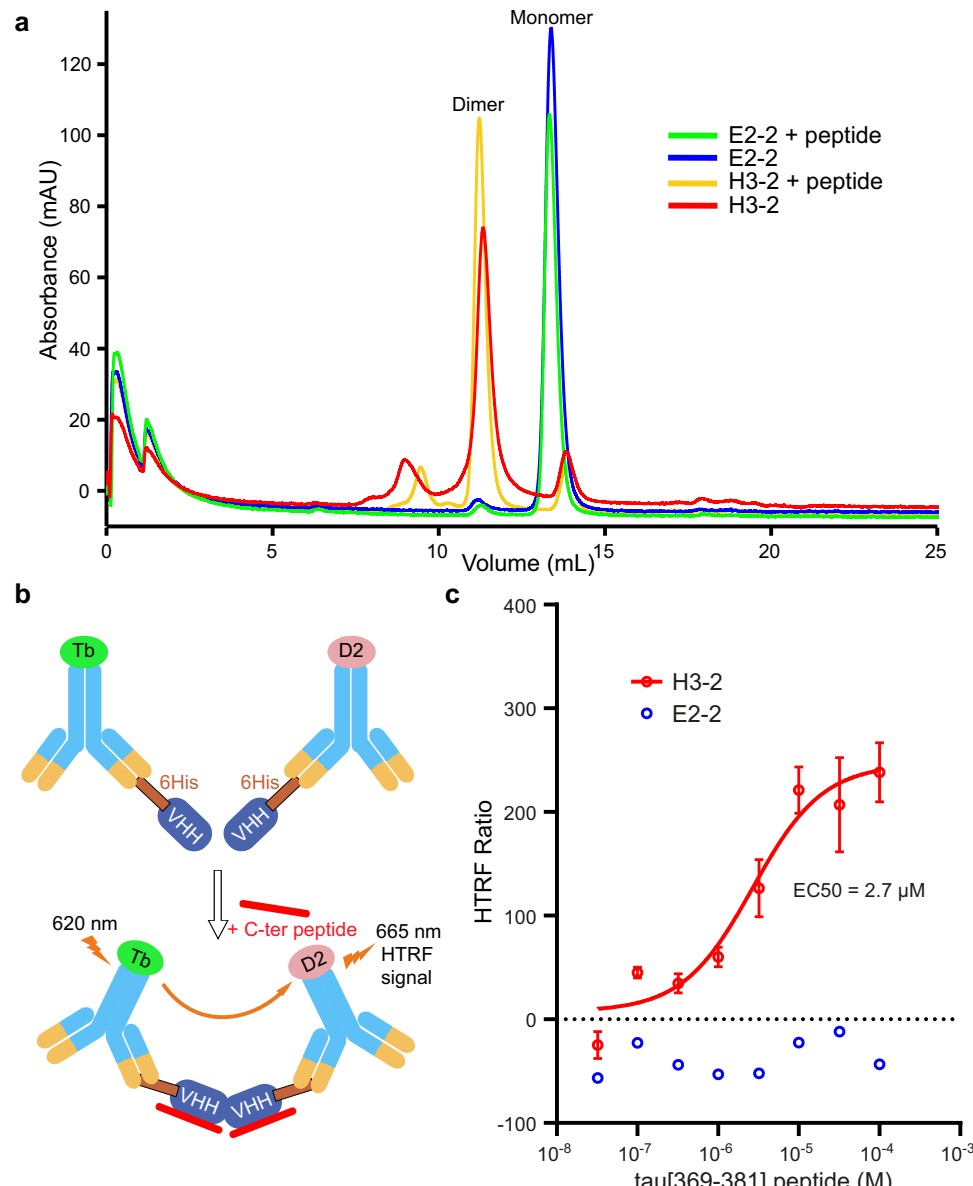

**Fig. 7 | VHH H3-2 spontaneously dimerizes in solution and its dimerization is stabilized by binding to tau[369–381] C-terminal peptide. a** Size exclusion chromatography (SEC) analysis ($A_{280 nm}$) performed on either VHH E2-2 (blue curve) and VHH H3-2 (red curve) alone, or in the presence of a 2-fold excess of the tau[369–381] C-terminal peptide (ratio 1: 2, green and yellow curves, respectively). Elution peaks corresponding to VHH monomer and dimer species are highlighted on the chromatogram (chromatogram is representative of a $n = 2$ independent experiments). Estimation of the VHH molecular weights was performed using a calibration curve obtained from SEC analysis of proteins of known size (Supplementary Fig. 15). **b** Schematic representation of the homogeneous time-resolved fluorescence (HTRF) assay performed with VHHs H3-2 and E2-2. Terbium (Tb) and D2 fluorophores correspond to the donor and acceptor, respectively. **c** Plotted HTRF ratio data corresponding to the concentration-response curve of the tau[369–381] C-terminal peptide in the presence of 6Histag-H3-2: donor and 6Histag-H3-2: acceptor (red dots, $n = 3$ independent experiments) or 6Histag-E2-2: donor and 6Histag-E2-2: acceptor (blue dots, $n = 2$ independent experiments). Data are presented as mean ± standard error of the mean for VHH H3-2 or mean only for VHH E2-2. An increase in the HTRF signal ratio is observed when the peptide concentration is increased in the presence of VHH H3-2 paired fluorophores. EC50 mean value was obtained using the nonlinear fit ([ligand] vs. response, 3 parameters) model.

dimerization can be enhanced through binding to the tau[369–381] C-terminal peptide. NMR competition assays in the presence of $^{15}$N-tau, VHH H3-2, and tauMTBD, at a molar ratio of 1: 1: 4, showed no modification of the pattern of $^{15}$N-tau resonance perturbations (Supplementary Fig. 13), confirming the absence of direct binding between VHH H3-2 and the MTBD, also shown by the SPR experiments (Supplementary Fig. 8). Alternatively, VHH H3-2 binding could indirectly disrupt the interaction of the LRP1 receptor with the tauMTBD by an effect on the conformational and/or dynamic rearrangement of the tau protein ensemble. This hypothesis may be supported by the very broad effect of VHH H3-2 binding on tau, as NMR resonance

broadenings were observed for residues located further from its peptide binding site, well into the MTBD, in stark contrast to VHH E2-2, which showed only a limited set of resonance broadenings corresponding to residues located close to the binding site (Supplementary Fig. 2 and Supplementary Fig. 13). Interestingly, the VHH H3-2 epitope is also accessible in the recombinant heparin-induced tau2N4R fibrils, as shown by their cryo-EM structures[49], which is not the case for the R2 and R3 repeat regions, targeted by the VHHs A31 and Z70$_{mut1}$, respectively. Consistently, we have shown that VHH H3-2 interacts with the recombinant tau fibrils, which may explain its selective activity in preventing the cellular uptake of tau fibrils compared to the

tauMTBD-binding VHHs. We have shown that tau fibril uptake is mediated by the HSPGs in the primary mouse neuronal culture (Supplementary Fig. 3 and Supplementary Fig. 10). Indeed, the VHH H3-2 binding site overlaps with tau-heparin binding sites as shown by in vitro NMR assays with monomeric $^{15}$N-tau, suggesting that its inhibition of tau fibril uptake may be due to competition with the HSPG binding[50].

The unique structural properties of VHH H3-2 proved to be of interest. The engineering of multivalent VHHs by "growing" scaffolds to improve affinity and increase molecular weight for multiple applications is an active field of research, be it the development of diabodies, the trimerization of VHHs for therapeutic use or the construction of "Legobodies" for cryogenic electron microscopy research[51–54]. Although the de novo design of a swapped dimer VHH may prove challenging, it demonstrates that the full potential of VHHs may yet be unlocked.

Here, we have demonstrated the interest and efficacy of tau-binding VHHs in blocking cellular uptake of tau. Thus, this series of VHHs represents an opportunity to better explore the epitopes and mechanisms of interest for targeting tau propagation, which are critical parameters for designing the best approach to tau immunotherapy[55,56]. In fact, the therapeutic antibodies may have different modes of action, as reflected by their individual efficacy in human clinical trials. To date, four anti-tau immunotherapies clinical trials (Phase II or III) have been discontinued due to lack of target engagement or efficacy[41]. Thus, tau immunotherapy requires the support of basic research and development to be properly designed and may benefit from the use of tau-binding VHHs to achieve this goal.

## Methods

### Ethics
The study was performed in accordance with the ethical standards laid down in the 1964 Declaration of Helsinki and its later amendments. The experimental research was performed with the approval of an ethics committee (agreement APAFIS #43474-2023050714441306 v6) and follows European guidelines for the use of animals.

### Selection and Screening of the VHHs Directed against tau protein
The Nali-H1 library of VHHs was screened against the recombinant biotinylated tau[57]. Briefly, recombinant tau protein was biotinylated using EZ-Link Sulfo-NHS-Biotin (Thermo Fisher Scientific), and biotinylated tau protein was bound to Dynabeads M-280 streptavidin (Invitrogen) at a concentration that was gradually decreased with each round of selection (100 nM, 50 nM, and 10 nM). Non-absorbed phage ELISA assay using avidin plates and biotinylated tau antigen (5 µg/ml) was used as a second step of cross-validation. VHH Z70$_{mut1}$ was obtained from an additional screen of optimization, using yeast two-hybrid method with tau protein as bait, following a strategy described in ref. 30. VHH A31 was obtained from a separate phage display screening of the Nali-H1 library, followed by a yeast two-hybrid-based optimization step as described[29,30]. Some of the VHHs used in this study are available at https://www.addgene.org/Isabelle_Landrieu/.

### Production and purification of VHHs
The VHHs were produced using a pET22b plasmid containing a recombinant sequence encoding a pelB leader sequence, a 6-His tag, a tobacco etch virus (TEV) protease cleavage site, and the desired VHH with or without an additional C-terminal cysteine[30]. Recombinant *E. coli* cells produced proteins targeted to the periplasm after induction by 1 mM IPTG in a terrific broth medium. Production was continued for 4 h at 28 °C or overnight at 16 °C before centrifugation to collect the cell pellet. The pellet was resuspended in 200 mM Tris-HCl, 500 mM sucrose, 0.5 mM EDTA, pH 8, and incubated on ice for 30 min. The resuspended pellet was then diluted four times in water, yielding

final concentrations of 50 mM Tris-HCl, 125 mM sucrose, 0.125 mM EDTA, pH 8, and complete protease inhibitor (Roche), and incubation was continued on ice for another 30 min. After centrifugation, the supernatant corresponding to the periplasmic extract was recovered. The VHHs were purified by immobilized-metal affinity chromatography (IMAC) (HisTrap HP, 1 ml, Cytiva), followed by SEC (Hiload 16/60, Superdex 75, prep grade, Cytiva) in phosphate buffer (50 mM sodium phosphate buffer [NaPi] pH 6.7, 30 mM NaCl, 2.5 mM EDTA, 1 mM DTT). VHHs were dialyzed against 50 mM Tris pH 8, 50 mM NaCl and cleaved with His-tagged Tobacco Etch virus (TEV) protease. The TEV protease and the cleaved 6-His tag were removed by a second IMAC step, and the VHHs were recovered in the flow-through, concentrated, and flash-frozen for further use.

### Production and purification of tau2N4R, tau1N4R, and tauMTBD fragment
pET15b-tau2N4R, tau1N4R, or tauMTBD recombinant T7 lac expression plasmids were transformed into competent *E. coli* BL21 (DE3) bacterial cells, and a small-scale culture was grown in LB medium at 37 °C. For production of recombinant $^{15}$N-tau2N4R, the small-scale culture was added to 1 L of a modified M9 medium containing MEM vitamin mix 1 × (Sigma-Aldrich), 4 g of glucose, 1 g of $^{15}$N-NH$_4$Cl (Sigma-Aldrich), 0.5 g of $^{15}$N-enriched Isogro Growth Powder (Sigma-Aldrich), 0.1 mM CaCl$_2$, and 2 mM MgSO$_4$. To produce other recombinant tau constructs, the small-scale culture was added to 1 L of LB medium at 37 °C.

Protein production was induced with 0.5 mM isopropyl-β-D-thiogalactopyranoside (IPTG) when the culture reached an optical density at 600 nm of 0.8. Cells were lysed in 50 mM NaPi pH 6.5, 2.5 mM EDTA, complete protease inhibitor cocktail (Sigma-Aldrich), and the tau proteins were first purified by heating the bacterial extract for 15 min at 75 °C. After centrifugation, the resulting supernatant was then passed on a cation exchange chromatography column (Hitrap SP Sepharose FF, 5 ml, Cytiva) equilibrated in 50 mM NaPi pH 6.5 and eluted with a NaCl gradient. Tau proteins were buffer-exchanged with 50 mM ammonium bicarbonate (Hiload 16/60 desalting column, Cytiva) before lyophilization. Detailed procedures are provided in refs. 58,59.

### NMR spectroscopy experiments
Analysis of the $^{15}$N-tau/VHH interactions was performed at 298 K on a Bruker Avance Neo 900.23 MHz spectrometer equipped with a cryogenic probe (CP-TCI [HCN] 5 mm Z-gradient). Trimethylsilyl propionate was used as an internal proton reference. Lyophilized $^{15}$N tau was dissolved in NMR buffer (50 mM NaPi, 30 mM NaCl, 2.5 mM EDTA, 1 mM DTT pH 6.8) containing 10% D$_2$O and mixed with a VHH at a final concentration of 100 µM for each protein. Two hundred microliters of each mixture in 3 mm tubes were sufficient to obtain the 2D $^1$H, $^{15}$N heteronuclear single quantum coherence (HSQC, Bruker hsqcetdpf3gpsi2) spectra with 32 scans. $^1$H, $^{15}$N HSQC were acquired with 3072 and 416 points in the direct and indirect dimensions, for 12.6 and 25 ppm spectral windows, in the $^1$H and $^{15}$N dimensions, respectively. The data were processed with Bruker Topspin 3.6 (https://www.bruker.com/en/products-and-solutions/mr/nmr-software/top.spin.html) and analyzed with POKY (poky.clas.ucdenver.edu)[60]. The spectra were rendered using POKY, and the intensity plots were designed using matplotlib[61].

### Characterization of the interactions of the full length tau2N4R, tau1N4R and tauMTBD with the VHHs
Affinity measurements were performed at 25 °C on a BIAcore T200 optical biosensor instrument (Cytiva). Recombinant tau2N4R and tau1N4R proteins were biotinylated with 3 molar excess of N-hydroxysuccimide (NHS)-biotin conjugates (Thermo Fisher Scientific) for 4 h at 4 °C. Recombinant tauMTBD fragment was biotinylated with (NHS)-biotin conjugates using a sub-stoichiometric ratio of 1: 0.5, for 4 h at 4 °C. Capture of biotinylated tau2N4R, tau1N4R and tauMTBD

in HBS EP+ buffer (Cytiva) was performed on a SA sensor chip (Cytiva), until the total amount of captured proteins reached 600, 500 and 100 resonance units (RU), respectively. A flow cell was used as a reference to assess non-specific binding and to allow for background correction. VHHs were injected sequentially at increasing concentrations ranging between 0.0625 and 4 μM in a single-cycle at a flow rate of 30 μL/min, with regeneration (three consecutive 1 M NaCl washes) between each VHH. Single-cycle kinetic analysis[62] was performed to determine association $k_{on}$, dissociation $k_{off}$ rate constants, and dissociation equilibrium $K_D$ constants by curve fitting of the sensorgrams using the 1:1 Langmuir interaction model of the BIA evaluation software 2.0 (Cytiva).

### Characterization of the interactions of VHHs with tau epitope-derived peptides

Affinity measurements were performed at 25 °C on a BIAcore T200 optical biosensor instrument (Cytiva). Recombinant VHHs H3-2 and E2-2 with a cysteine at the C-terminus were biotinylated with 20 molar excess of maleimide biotin conjugates (Thermo Fisher Scientific) overnight at 4 °C. Capture of biotinylated VHHs was performed on a SA sensor chip in HBS EP + as for tau immobilization. A flow cell was used as a reference to assess non-specific binding for background correction. Biotinylated VHHs H3-2 and E2-2 were injected onto a SA chip until the total amount of captured proteins reached 660 and 210 RU, respectively. 9 consecutive and increasing ½ log concentrations of tau-Cter peptide $_{369}$KKIETHKLTFREN$_{381}$ (also named tau[369–381]) were injected sequentially at a flow rate of 30 μL/min, in a multi-cycle, with regeneration (one wash of 1 M NaCl between each peptide injection). Multi-cycle kinetic analysis was performed to determine association $k_{on}$, dissociation $k_{off}$ rate constants, and dissociation equilibrium $K_D$ constants by curve fitting of the sensorgrams using the 1:1 Langmuir interaction model of the BIA evaluation software 2.0 (Cytiva). $K_D$ determination was also obtained by plotting the individual response value to each concentration of tau-Cter peptide. Equilibrium data were fitted to a binding model using non-linear regression analysis (one binding site, specific) available in GraphPad Prism software. The stoichiometric ratio (SR) value is the ratio between the observed maximum binding response to the theoretical maximum binding response (Theo Rmax). Theoretical Rmax is given by the molecular weight (MW) ratio of the analyte to the ligand, multiplied by the expected stoichiometry, and the amount of immobilized ligand (in RU), as follow:

$$Theo\,R\max = Immobilized\,VHH\,Amount\,(RU)\,\frac{MW\,peptide}{MW\,VHH}. \qquad (1)$$

### Fluorescent labeling of recombinant tau1N4R and tauMTBD fragment

50 μM of recombinant tau1N4R in PBS buffer (Euromedex) was incubated with 150 μM of Alexa Fluor™ 546 C5-maleimide (Invitrogen) to attach the dye to its two native cysteine residues for 4 h at 4 °C. The reaction was quenched with 2 mM DTT. 50 μM of recombinant tau1N4R in MES buffer (50 mM MES pH 6.9, 30 mM NaCl, 2.5 mM EDTA) was incubated with 150 μM of Atto565 NHS ester (Invitrogen) for 4 h at 4 °C. 50 μM of recombinant tauMTBD in PBS buffer (Euromedex) was incubated with 50 μM of Atto488 NHS ester (Sigma) for 4 h at 4 °C. Reactions were quenched with 10 mM Tris-HCl pH 7.4. Residual probes were removed by two consecutive buffer exchanges using a Zeba spin desalting column.

### Preparation of labeled tau1N4R and tau2N4R fibrils

Labeled tau1N4R (Atto565) or tau2N4R fibrillization were performed with 10 μM of tau in MES buffer, supplemented with 0.3 mM DTT and 2.5 μM heparin H3 (Sigma-Aldrich), at 37 °C for 72 h. Tau1N4R fibrils

used for the tau uptake assays were sonicated for 1 min using a sonicator bath, aliquoted, and flash-frozen for further use.

### Transmission electron microscopy experiments

3 μL samples of Atto565-labeled tau1N4R fibrils or unlabeled tau2N4R fibrils were loaded onto a formvar/carbon-coated grids, incubated for 5 min, and then the grids were rinsed three times with PBS. For VHH H3-2 gold immunostaining, the grids were blocked with 1 % BSA in PBS for 15 min. VHH H3-2 was coated with 10 nm gold beads according to the manufacturer's protocols (Gold Conjugation Kit (10 nm, 20 OD), Abcam), and 3 μl of a 1/20 dilution of the final product was applied to the grid, incubated for 1 h, and then the grid was washed 5 times with PBS. The grids were stained with 2% uranyl acetate in 3 consecutive steps of 10 seconds, 1 s, and 1 min incubations before final drying. Tau fibrils and VHH H3-2 immunostaining on fibrils were observed under a JEOL JEM-2100 transmission electron microscope, and images were captured with a Gatan Orius SC200D camera.

### Primary neuronal culture

Mouse primary cortical cells were prepared from 15 days-old C57BL/6JRj mouse embryos, as previously described, in a sterile class II biosafety cabinet (Thermo Scientific PSM HERASAFE KS)[63]. C57BL/6JRj gestating mice (Janvier Labs Le Genest-Saint-Isle, FRANCE) were housed in a temperature-controlled (20–22 °C) room with a 12 h light, 12 h dark cycle. Food and water were provided *ad libitum*. The culture medium was made of Neurobasal (Gibco) supplemented with B-27 (Gibco), Antibiotic-Antimycotic (Gibco), and L-glutamine (Gibco). The number of viable cells in the cell suspensions was determined through the trypan blue exclusion test (Gibco). 300k cells were plated in poly-D-lysine (0.5 mg/ml, Sigma) and laminin (10 μg/ml, Sigma)-coated 24-well plates (Falcon). Cultures were maintained in a cell incubator (Thermo Fisher Scientific) at 37 °C in a humid atmosphere with 5% of $CO_2$.

### LDH assay

To assay cell death, the release of lactate dehydrogenase (LDH) into the cell medium was quantified using the CytoTox 96 non-radioactive cytotoxicity assay (Promega) following the manufacturer's instructions. Maximum LDH release, defined as 100% of toxicity, was determined by adding lysis solution (9% Triton X-100) to the cells.

### Tau cellular uptake assays

Experiments were performed on primary neuronal cultures at DIV 20 (differentiation day 20). 50 nM final concentration of Alexa546-labeled monomeric tau1N4R, 200 nM of Atto565-labeled tau1N4R fibrils or 500 nM of Atto488-labeled tauMTBD was co-incubated with either one of the anti-tau VHHs at different concentrations (30, 100, 300, 400 or 1000 nM) or just PBS (tau_M), for 30 min at room temperature. Positive controls for cellular uptake inhibition consisted of incubation of tau 1N4R or tau 1N4R fibrils with 100–400 nM of human RAP protein (Enzo Life Science) or 50 μg/mL low molecular weight heparin (AMSBIO), respectively, while negative controls consisted of incubation with 100, 400 or 1000 nM of a VHH targeting the green fluorescent protein (GFP). The tau-VHH or tau-control mixtures were then diluted into the medium of primary neuronal cultures and further incubated at 37 °C. We did not observe any cellular toxicity after treatment with the different tau-VHH complexes based on a lactate dehydrogenase (LDH) assay, at any assayed concentration or for any form of tau (Supplementary Fig. 16a–e). After 1 h, tau uptake was assessed based on the intracellular fluorescence signal using a fluorescence readout after cell lysis or by confocal microscopy. The cells were washed twice in pre-warmed PBS, treated with 0.01% trypsin for 2 min at 37 °C, and washed again twice with PBS.

For measurement of the intracellular fluorescence signal, washed cells were lysed by adding 200 μL of RIPA buffer (Thermo Fisher Scientific) supplemented with protease inhibitor cOmplete™ EDTA free

(Roche), Benzonase® Nuclease (Merck) (1U/μL final) and then vortexed. 150 μL of lysates were transferred to a Nunc™ flat-bottom 96-well microplate (Thermo Fischer Scientific). Alexa546, Atto565, or Atto488 fluorescence intensity was recorded using a Varioskan™ LUX multi-mode microplate reader (Thermo Fischer Scientific). Intracellular tau fluorescence data were normalized to the fluorescence of monomeric tau1N4R, tau1N4R fibrils, or tauMTBD in PBS conditions.

For confocal microscopy analysis, the cells were washed twice in pre-warmed PBS and treated with 0.01% trypsin for 2 min at 37 °C. The cells were fixed with 4% paraformaldehyde for 30 min at room temperature. After three consecutive washes in PBS, the nuclei were stained with 40,6-diamidino-2-phenylindole (DAPI, 1/10 000) for 15 min at room temperature. Cells were cover-slipped with DAKO, and images were captured using a Zeiss AxioObserver Z1 (Yokogawa CSU-X1 spinning disk, sCMOS Photometrics Prime 95B camera). Images have been processed using ZEN software (Zeiss).

### Competition assay between tau1N4R and the anti-tau VHHs on LRP1 cluster III domain

Competition assays were performed with a BIAcore T200 optical bio-sensor instrument (Cytiva). Recombinant Human LRP1 cluster III protein, Fc Tag (ACROBiosystems AG), was captured on a Serie S Protein G sensor chip (Cytiva) surface to an 1800 RU level, at a flow rate of 10 μL/min and using a working solution of 12 μg/mL in HBS-N buffer (Cytiva) supplemented with 0.005% surfactant P20 (Cytiva) and 1 mM CaCl$_2$ final concentration. The same supplemented buffer was used for binding and competition assay experiments. One regeneration step was performed following each cycle of binding or competition step using 10 mM Glycine-HCl pH 1.5 (Cytiva) at a flow rate of 10 μL/min. Recombinant tau1N4R was injected sequentially at increasing concentrations ranging between 0.012 and 1 μM (from 0.5 log concentration dilutions) in a single cycle. Single-cycle kinetic analysis[62] was performed to determine association $k_{on}$, dissociation $k_{off}$ rate constants, and dissociation equilibrium $K_D$ constant by curve fitting of the sensorgrams using the 1:1 Langmuir interaction model of the BIA evaluation software 2.0 (Cytiva). For the competition assays, 100 nM of recombinant tau1N4R was co-injected in the absence or presence of increasing concentrations of VHHs (30, 100, and 300 nM) at a flow rate of 30 μL/min, on immobilized LRP1 cluster III protein. Blank corrections were made by subtracting the response curves of VHHs injections without tau from the corresponding tau-VHHs co-injection response curves. A positive control for inhibition of the tau-LRP1 cluster III interaction consisted in the co-injection of tau1N4R in the presence of human RAP protein (Enzo Life Science) at a final concentration of 300 nM.

### VHH H3-2 crystallization and structure determination

H3-2 concentrated to 280 μM was incubated with 1 mM of tau-Cter peptide for 30 min prior to crystallization screening. From an initial screening of approximately 600 conditions, the optimal crystallization condition was found to be 23% PEG-MME 2000 and 0.1 M potassium acetate (derived from a condition found in the JCSG + Suite, Qiagen). Crystals were cryoprotected using 10 % glycerol within the crystallization condition. Crystals were analyzed at the SOLEIL synchrotron (Paris, Fr) beamlines PX1 and PX2A. Crystals belonged to space group P2$_1$2$_1$2$_1$ with cell parameters suggesting that the asymmetric unit contains 4 monomers (42% probability estimated from the Matthews coefficient). The best diffraction dataset was obtained at a resolution of 1.8 Å. The structure was solved by molecular replacement (MOLREP) with pdb 7qcq as a template and refined to a Rwork of 0.18 and Rfree of 0.23, using REFMAC5 and COOT (Supplementary Table 1)[64–66]. The structures have been deposited in the Protein Data Bank (PDB) under the accession code 9G13. The structure images were created using PyMOL Molecular Graphics System (Version 1.2r3pre, Schrödinger, LLC).

### Size exclusion chromatography experiments

Size exclusion chromatography (SEC) analyses were performed with an ÄKTA Pure FPLC system (Cytiva) equipped with a multi-wavelength detector using a Superdex™ 75 Increase 10/300 GL column (Cytiva) equilibrated in PBS buffer (Euromedex), with a flow rate of 500 μL/min at 16 °C. A loop of 100 μL was used to inject the samples. The elution of 20 μM of VHH E2-2 or VHH H3-2 was analyzed in the absence or presence of 40 μM of tau-Cter peptide. Recombinant proteins were prepared in the PBS buffer and were mixed with peptide prior to injection into the column. Molecular weights (MW) were calibrated using a gel filtration Calibration Kit – Low Molecular Weight (Cytiva).

### HTRF VHH dimerization assay

The HTRF assay was performed in white OptiPlate-96 microplates (PerkinElmer) with a total volume of 20 μL per well. 6His-VHHs were premixed with anti-6His antibodies (Ab) conjugated with fluorescence donor or acceptor at a 2:1 (Anti-6His Ab: 6His-VHH) ratio for 10 min at room temperature. 5 μL of 6His-VHH/anti-6His Ab Terbium (fluorescence donor 620 nm, final concentration 1.5 nM) and 5 μL of 6His-VHH/anti-6His Ab D2 (fluorescence acceptor 665 nm, final concentration 20 nM in the Terbium detection buffer (Revvity)) were added to the plate. Then 10 μL of 8 increasing 0.5 log concentrations of tau-Cter peptide, with final concentrations ranging from 32 nM to 100 μM, were added to the mixture and incubated for 1 h at room temperature in the dark. HTRF measurements were recorded using a Varioskan™ LUX multimode microplate reader (Thermo Fischer Scientific) for optimal signal detection at 620 nm and 665 nm. The 665/620 nm ratio can measure the combination of two VHHs in the presence of the tau-Cter peptide. The individual HTRF ratio value against each concentration of tau-Cter peptide was fitted to a binding model using non-linear regression analysis ([ligand] vs. response, 3 parameters) available in GraphPad Prism software.

### Statistical analysis

Data are presented as the mean ± standard deviation (SD) for SPR assays. Data are presented as the mean ± standard error of the mean (SEM) for HTRF assays. Tau cellular uptake and LDH assays are presented as boxplots, and normalized values are given as mean ± standard deviation (SD). Ordinary one-way nonparametric ANOVA (Kruskal-Wallis) with Dunn's multiple comparison tests was used to analyze data for monomeric tau1N4R, tau1N4R fibrils, and tauMTBD cellular uptake experiments.

### Reporting summary

Further information on research design is available in the Nature Portfolio Reporting Summary linked to this article.

## Data availability

Crystallography data generated in this study are deposited in the Protein Data Bank under accession code 9G13. All other processed data generated in this study are available in the Supplementary Information and the Source Data file. Source data is made available on Figshare [https://doi.org/10.6084/m9.figshare.28234610].

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

## Acknowledgements

We thank Ms M. Oosterlynck and S. Bégard for their help with the preparation of primary neuronal cultures. We thank M. P. Legrand for valuable help with data collection at beamline PX1 of the SOLEIL synchrotron facility (Paris, France). We thank SOLEIL for providing the synchrotron radiation facilities. We also thank the BiCel platform of US 41 - UAR 2014 - PLBS for the microscopy data acquisition and M.L. BRUNET for access to instruments and technical assistance for the TEM observations performed on instruments purchased with funds from the Agence Nationale de la Recherche: ANR 10-EQPX-0004. The NMR facilities were funded by the Conseil Régional du Nord, the CNRS, the Institut Pasteur de Lille, the European Union, the French Ministry of Research, and the University of Lille. The financial support of the IR INFRANALYTICS FR2054 CNRS for the conduct of the research is gratefully acknowledged (I.L.). We acknowledge financial support from the ANR (Agence Nationale de la Recherche, France), projects ANR-11-LABX-01 (L.B. and I.L.), ANR-18-CE44-0016 (L.B. and I.L.) and ANR-22-CE92-0061 (L.B. and I.L.). This project was partially funded by the European Union (HORIZON-MSCA-2022-DN-01, Project TAME, GA101119596, I.L.), by the Rainwater Charitable Foundation & Alzheimer's Association (Project T-PEP-23-969176, L.B.) and the France Alzheimer's Association (Project 6433, C.D.).

## Author contributions

C.D., E.D., T.B., M.D., C.L., M.N., J.M., and F.-X.C. investigation; C.D., E.D., J.-C.R., X.H., M.C., L.B., and I.L. methodology; C.D., E.D., T.B., M.D., and M.C. validation; C.D. and E.D. visualization; C.D., E.D., M.C., L.B. and I.L. supervision; C.D., E.D., L.B., and I.L. conceptualization; C.D. and E.D. formal analysis; L.B. and I.L. project administration; C.D., L.B., and I.L. funding acquisition; C.D. and I.L. writing-original draft. C.D., E.D., J.-C.R., X.H., M.C., L.B., and I.L. writing-review and editing.

## Competing interests

J.-C.R. is the CEO of Hybrigenic Services. C.D., E.D., J.-C. R., L.B., and I.L. are the inventors of a patent (WO2020/120644A1) that covers the use of VHH Z70 and VHHs derived from it. The remaining authors declare no competing interests.
