## [Transparent Peer Review file · Nature Communications]

Inhibition of tau neuronal internalization using anti-tau single domain antibodies

Corresponding Author: Dr Isabelle Landrieu

Version 0:

Reviewer comments:

Reviewer #1

(Remarks to the Author)

Preliminary comments on "Inhibition of tau neuronal internalization using anti-tau single domain antibodies" by Danis et al (Nature Comm manuscript)

This manuscript by Danis et al., the authors generated and evaluated two VHH nanobodies (A31, Z70mut1) that bind to the microtubule binding domain of tau for uptake inhibition of tau into cells. They also provide evidence that a third VHH (H3-2) forms a dimer upon binding to C-terminal tau. The in vitro and structural data on generation and characterization of the VHHs is convincing. The quality of the uptake assays in cells is somewhat limited and would benefit from additional experiments as outlined below. One major shortcoming of the in vitro and cell experiments is that the study is limited to tau monomer, and tau aggregates were not tested. The observation that a VHH can dimerize when interacting with tau is novel and potentially important for the field since this mechanism could enhance the potency of VHHs as uptake inhibitors. Overall, this study is of potential interest to the readers of Nature Communications. I recommend major revisions and re-submission.

Major comments

Introduction

Page 2, 1st paragraphs: The authors discuss LRP1 however not HSPGs as uptake receptors for tau. There is an extensive body of literature in the field on tau uptake via HSPGs which should be mentioned here.

Methods

Page 7, tau cellular uptake assay:

- o The concentrations of tau are non-physiologically high (500 nM). Lower tau concentrations should be tested. Likewise, since fibrils are presumably the target in disease, they must be tested as well.
- o Could the VHH concentrations in the micromolar range realistically be achieved in vivo?
- o The authors would benefit by using lower (more physiologic) concentrations of tau and tau fibrils, which would necessitate lower inhibitory VHH concentrations, and might improve their assays.
- o To measure uptake by lysing the cells with RIPA may change/ quench the fluorescence and cause significant variability across replicates, and may mistakenly measure tau monomer trapped on the cell surface. Flow cytometry (after quenching/trypsinization) based on fluorescence in intact cells, or confocal microscopy, should be used to cross-validate the findings.
- o Other known inhibitors of tau uptake should be used as positive controls, e.g., heparin (for HSPGs) and RAP (for LRP1), especially to help us understand the magnitude of the effects.
- o Why did the authors choose the tau isoform 1N4R over other isoforms for uptake assays? Do the VHH prevent uptake of other isoforms as well?

Results

Page 6, 1st paragraph: How did the authors select for the nanobodies? We understand that this work was done in prior publications. It would be helpful to include a brief summary in the result section or method section.

Page 13: see comments on uptake assay with lysis above

Page 15, 2nd paragraph: The authors incorrectly state that LRP1 mainly mediates tau uptake. Multiple papers in the literature show that other receptors (including HSPGs) play a significant role as well. Please also correct similar statements on page 19.

Page 16, top of page: The authors report: "VHH H3-2, which showed the strongest inhibition of cellular tau uptake, was also tested but strong unspecific binding on the chip prevented valid measurements." Could the authors comment in more detail what is meant with this statement and why there was non-specific binding?

Page 20, middle paragraph:

- The authors demonstrate binding of tau 1N4R to cluster III domain of LRP1 in vivo, and provide in vitro data for competition of A31 and Z70mut1 with the tau:LRP1 cluster III interaction. We would suggest confirming this result in vivo, e.g., stain for LRP1 on the cell surface and then assess for co-localization with tau in the presence and absence of VHH.

- Given the relevance of other receptors for tau uptake, it would greatly increase the impact of this paper if the authors checked for competition of the VHS with tau:HSPG binding.

Page 21, 1st and 2nd paragraph: the observation that H3-2 forms dimers that are enhanced by C-terminal tau binding is interesting. I am wondering if H3-2 somehow promotes fibrillization of tau (this is true for heparin, which is also used as a very effective uptake inhibitor of tau). The authors could test this easily by incubating H3-2 with tau monomer and checking with Thioflavin T and TEM for aggregation.

Figures

Figure 1: None of the selected VHH studies in this paper bind the N terminus, which is reportedly the binding site to the cell membrane. Could the authors comment on this?

Figure 2B: The experimental variance for these uptake assays is quite large. This could be optimized by using flow cytometry or confocal microscopy to quantify fluorescence in intact cells. How many replicates were done per condition in each experimental run? Please include this information in the legend.

Figure 2C and 3C: The quality of the confocal images is poor. It is hard to tell if labeled tau is inside of cells or not, although fluorescence is significantly reduced for A31, Z70mut1 and H3-2. Please provide higher quality images.

Minor comments

- Introduction page 2, 1st paragraph: the majority of statements made here are missing references. Please include appropriate references.

- Please correct spelling/ grammar mistakes

Reviewer #2

(Remarks to the Author)

Reviewer #3

(Remarks to the Author)

In their work the authors investigate different single domain anti-tau antibodies or VHHs, targeting a series of different epitops of the free soluble tau to inhibit neuronal internalization. They show which of the used VHHs is the most potent inhibitor by affinity and cellular uptake measurements. Finally, using a combination of biophysical methods, they identify VHH H3-2 as the strongest inhibitor for cellular tau uptake. The new VHHs may be useful as an immunological tool in the studies of prion-like propagation tauopathies.

The work addresses a timely and relevant problem, the manuscript is well structured and the main aims are clearly explained. The methods are clearly documented. However, further experiments are required to confirm and justify the authors' claims and conclusions before the work may be considered for publication in Nature Communications.

Specific remarks:

1.) In the NMR and affinity experiments the authors used tau2N4R, whereas in the cellular uptake and LRP1 receptor-binding experiments they instead used tau1N2R. The authors should explain why they used different tau constructs and what implications this may have had?

2.) SI Fig. 2B: It looks like the concentration of the sample with tau-VHH was higher than of the reference tau. Could this affect the results of much higher intensities than 1? This does not seem to be observed in such an extent in the other spectra.

Please verify.

3.) SI Fig. 2A and SI Fig. 4A: In both spectra tau degradation is observed. Why did tau degradation occur and how could/does this affect the results? This is not clear.

4.) line 333: SCK experiments confirmed the binding ...

The SCK experiments show quite different affinities of VHH A31 and Z70mut1 when using the MTBD only. The value of the K_d of A31 to tau was 116 nM compared to 41 nM to tauMTBD, but the striking difference is the affinity of Z70mut1 to tau 21 nM compared to 799 nM to tauMTBD. The authors should explain this 38-fold difference in the K_d of Z70mut1 and its influence on the binding/function. In addition, the authors write that no interaction between tauMTBD and VHH H3-2 was detected, even though the NMR binding profile shows prominent signal attenuations in the repeat region as well. The authors should comment on this.

5.) In the binding experiment to cluster III of the LRP1, VHH A31 and Z70mut1 together with tau were measured but not H3-2, which showed the strongest inhibition of cellular uptake. What could be the cause of the unspecific binding of the VHH H3-2 to the chip? Could it be the formation of a dimer? Is it possible to try different concentrations or conditions in order to measure/detect H3-2 as well? In my opinion this is an important experiment that would provide evidence that may (or may not) justify the conclusions for H3-2 as well.

6.) line 506: Because of this specific binding mode, VHH H3-2 could indirectly disrupt the interaction of a LRP1 receptor with tauMTBD, ... Please provide additional experiments supporting this claim, so far it seems speculative.

Minor remark:

1.) line 307: tau uptake

2.) line 628: Please correct the title

3.) line 3 in SI: Please correct to Figure 1 (now Fig. 2)

4.) It would be informative to have the sequence alignment of all three C-ter binding VHH (in the SI Fig. 3).

5.) Please arrange the citation of the SI Figures in the manuscript in the order they appear in the Supplementary Material.

6.) SI Fig. 6. Please correct the title

Reviewer #4

(Remarks to the Author)

A very interesting study on inhibition of neuronal internalization of free soluble tau by VHH antibodies. While several VHH were found to bind to various epitopes in tau2N4R, VHH A31, Z70mut1 and H3-2 were found to strongly inhibit tau cellular internalization. Two of these also competed with the LRP1 receptor. Three VHH, F8-2, E2-2, and H3-2, were found to bind to the C-terminal domain with F8-2 binding more weakly in the μ M range and H3-2 more strongly in the low nM range. H3-2 was the most potent in inhibiting cellular tau uptake, with E2-2 being intermediate, whereas F8-2 was inactive. The inhibition was somewhat correlated with their relative epitope binding affinities. A crystal structure was obtained for VHH H3-2 with the tau C-terminal peptide (residues 369-381) and the VHH was found to unexpectedly form a domain-swapped dimer where the CDR H3 and FR4 were swapped with their counterparts in the VHH dimer and the CDR H3 formed an anti-parallel β -sheet. To rule out crystallization artifacts, the H3-2 VHH was analyzed in solution by size exclusion chromatography and was found as a predominantly dimeric form in the presence and absence of tau peptide compared to E2-2, which formed a monomer. Thus, there is supporting evidence for F8-2 forming a biological dimer. I will focus mainly on the structural work and have the following comments.

Page 10, line 229. Please subscript all of the 1's in P212121 to read P₂1₂1₂1.

Page 12, line 277. Please replace "complementary" with "complementarity".

Page 18, lines 417-418. "The interaction is mainly driven by main-chain hydrogen bonds between the peptide and the CDR3 in a parallel β -sheet conformation, forming a 4-stranded β -sheet with 2 CDR3 and 2 peptides". So where does the sequence specificity for this particular peptide arise if the main interactions are to the peptide backbone? There needs to be more discussion concerning this point. Do other peptides bind to this domain-swapped dimer? Does peptide also bind to the monomeric VHH?

Pages 21-22, lines 495-496. "The only difference between VHHs F8-2 and E2-2 are their CDR1 and CDR2 sequences, which only modulate their affinity for tau, not their epitope". Sorry I don't follow this. Is the epitope not part of tau? What are the CDR1 and CDR2 sequences recognizing - or does the CDR1 and 2 sequences only indirectly influence binding and epitope affinity.

Page 21, lines 499-500. "differences observed for the VHHs E2-2 and H3-2, because their affinities are within the same range, as shown by SPR" I don't know that I would agree with that. There is still a four to five fold difference in K_d, and 4 fold difference in the kon. What is the binding affinity of TauMTBD to receptor and are these different VHH biological activities related to that?

Page 32. There is no crystallographic table that I could find and, hence, the paper should not be accepted without it – sorry if

I missed it somehow but I also could not also see any reference to tables in the text. This table needs to be reviewed. There was however a PDB report but the Ramachandran outliers were in the red region (unfavorable) as well as the RSRZ outliers. I would expect that a 1.8 Å structure would not have such a poor Ramachandran outlier score without some explanation. The data and refinement statistics in the PDB stats are also incomplete. There are no values for Rmerge in the outer shell, no Rpim, no B-factors for tau peptide compared to the VHH or overall. These should be included in a data processing and refinement statistics SI table.

Page 34, Reference 1. Incomplete reference.

Page 35, Reference 21. Incomplete reference.

Page 37, Reference 41. Incomplete reference.

Page 37, Reference 46. Incomplete reference.

Page 38, Reference 53. Incomplete reference.

Page 38, Reference 54. Incomplete reference.

Version 1:

Reviewer comments:

Reviewer #1

(Remarks to the Author)

I am satisfied that the authors have addressed reviewer concerns sufficiently for the manuscript to be published.

Reviewer #2

(Remarks to the Author)

Reviewer #3

(Remarks to the Author)

The authors have addressed my questions.

Reviewer #4

(Remarks to the Author)

Almost all of my previous comments have been satisfactorily addressed in the revision. However, I have some other minor comments that stem from the revisions. It would have also been nice to have the page numbers and lines for the revisions to the text so not to have to search through the paper looking for them as the search function doesn't seem to work so well on PDF files.

Page 14. "The full table of statistics is now available as SI Table 1.

I have some comments on the SI Table 1

Resolution range. 38.17 - 1.8 (1.864 - 1.8)

Please replace first "1.8" with "1.80" and second with "1.800" if that is what it is so the number of decimal points are the same in each category? "

Unit cell. "70.394 76.346 95.565" Two decimal points are sufficient.

Mean I/sigma(I). "26.27 (3.40)". One decimal point is sufficient.

Wilson B-factor R-merge. Please add units after B-factor (Å²) and change the value to an integer - decimal points are not meaningful.

R-merge, R-meas, Rpim. 3 decimal points are sufficient.

R-work, R-free. 3 decimal points are sufficient.

Please replace "RMS(bonds), RMS(angles)" with "RMS Bond lengths (Å), RMS Bond angles (°)".

Please replace “Average B-factor” with “Average B-factor (A²)”.

macromolecules, solvent, VHHs, peptides. Please truncate all B values to integers.

Page 14, “All incomplete references have been corrected”.

However, not all are complete, see below.

Page 32, line 776, Reference 30. Please replace “300, (2024)” with “300, 107163 (2024)”.

Page 32, line 778, Reference 31. Please replace “eLife 5, (2016)” with “eLife, 5, e16228 (2016)”.

Page 34, line 834, Reference 57. Please replace “7,” with “7, e36584”.

Please find below our point by point rebuttal to the reviewer's comments.

Inserted text from the revised version is here in blue and is highlighted in yellow in the revised manuscript.

The reviewers comments are in red, our replies in black.

Reviewer #1 (Remarks to the Author):

This manuscript by Danis et al., the authors generated and evaluated two VHH nanobodies (A31, Z70mut1) that bind to the microtubule binding domain of tau for uptake inhibition of tau into cells. They also provide evidence that a third VHH (H3-2) forms a dimer upon binding to C-terminal tau. The in vitro and structural data on generation and characterization of the VHHs is convincing. The quality of the uptake assays in cells is somewhat limited and would benefit from additional experiments as outlined below. One major shortcoming of the in vitro and cell experiments is that the study is limited to tau monomer, and tau aggregates were not tested. The observation that a VHH can dimerize when interacting with tau is novel and potentially important for the field since this mechanism could enhance the potency of VHHs as uptake inhibitors. Overall, this study is of potential interest to the readers of Nature Communications. I recommend major revisions and re-submission.

Major comments

Introduction

Page 2, 1st paragraphs: The authors discuss LRP1 however not HSPGs as uptake receptors for tau. There is an extensive body of literature in the field on tau uptake via HSPGs which should be mentioned here.

The revised introduction now mentions the critical role of HSPGs in the uptake of oligomeric and fibrillar forms of tau, with the following sentence and associated references

"Micropinocytosis and active endocytosis have been described to mainly drive the cellular uptake of tau fibrils and it requires the involvement of heparan sulfate proteoglycan (HSPGs). 4–6"

10. Holmes, B. B. et al. Heparan sulfate proteoglycans mediate internalization and propagation of specific proteopathic seeds. *Proc. Natl. Acad. Sci. U. S. A.* (2013).

11. Rauch, J. N. et al. Tau Internalization is Regulated by 6-O Sulfation on Heparan Sulfate Proteoglycans (HSPGs). *Sci. Rep.* (2018).

12. Stopschinski, B. E. et al. Specific glycosaminoglycan chain length and sulfation patterns are required for cell uptake of tau versus α -synuclein and β -amyloid aggregates. *J. Biol. Chem.* (2018).

13. Marvian, A. T. *et al.* Distinct regulation of Tau Monomer and aggregate uptake and intracellular accumulation in human neurons. *Mol. Neurodegener.* **19**, 100 (2024).

And it is further discussed in the revised version, specifically in the view of the recent reference 13.

In addition, further experiments were performed with LMW heparin as suggested (See below for additional experiments with known inhibitors).

Methods

Page 7, tau cellular uptake assay:

o The concentrations of tau are non-physiologically high (500 nM). Lower tau concentrations should be tested. Likewise, since fibrils are presumably the target in disease, they must be tested as well.

To address this comment, we have first performed a concentration response curve of fluorescently pre-labeled monomeric tau and tau fibrils in the tau cellular uptake assay (1h incubation, 37°C, primary neuronal cultures), which is presented here:

We tested 50, 100, 200 and 500 nM of tau in both conditions (monomeric and fibrils). We have observed a linear response in this range and that 50 nM monomeric tau and 200 nM of tau fibrils gave a fluorescence signal (0.3) with a sufficient signal to noise to measure the activity of the VHHs.

We therefore next performed an additional neuronal uptake assay using a lower concentration of monomeric tau1N4R (50 nM) with 100 nM of each anti-tau VHH. Additionally, we also performed the uptake assay with 30 and 300 nM of the anti-tau VHHs A31, Z70mut1, E2-2 and H3-2, which are the VHHs of most interest in this study. The data are now included in revised Figure 2, SI Figure 7 and SI Figure 8.

The results obtained were consistent with our previous observation using the higher concentration of monomeric tau (500nM) and VHHs. VHH H3-2 showed the best inhibitory effect, followed by VHHs A31 and Z70mut1, the latter only showing significant activity at a concentration of 300 nM.

We also performed an additional tau neuronal uptake assay using 200 nM of recombinant tau1N4R fibrils in the presence of 400 nM of anti-tau VHHs. Interestingly, VHH H3-2 was the only VHH that significantly reduced tau fibrils uptake. This data is now included in revised Figure 4 and SI Figure 11.

We have also shown that VHH H3-2 binds to preformed tau fibrils *in vitro* using TEM (included in the revised version as SI Figure 12).

Finally, we have verified the absence of toxicity assay for all these new conditions. These data are now included in SI Figure 1.

o Could the VHH concentrations in the micromolar range realistically be achieved *in vivo*?

We have some factual elements to respond to this important comment.

- Firstly, we have now repeated the experiments at lower concentrations (down to 30 nM), showing that the most efficient VHHs are still effective in blocking cellular uptake, specifically VHH H3-2 (SI Figure 7). This demonstrated the potential for activity in the sub-micromolar range.

- We have also shown that both epitope and affinity are likely parameters to modulate the activity of VHHs. In terms of affinity, we have previously demonstrated our ability to engineer these VHHs to achieve higher potency (see Refs 28 and 29), opening the way for the development of some of these VHHs towards therapeutic application.

- The approach we have taken so far, in tauopathy mouse models, is to deliver the VHHs using AAV viral vectors, a gene therapy approach that is compatible with VHHs due to their small size. In this way, we have achieved quite high levels of local expression (in the hippocampus, for example see Ref 29). The

same concern applies to monoclonal IgG, only a small percentage of which crosses the BBB, but they are in clinical trials.

o The authors would benefit by using lower (more physiologic) concentrations of tau and tau fibrils, which would necessitate lower inhibitory VHH concentrations, and might improve their assays.

We have performed additional assays using the lowest possible concentrations of monomeric tau and tau fibrils that still give a signal to noise ratio compatible with the fluorescence measurements, as described above.

O To measure uptake by lysing the cells with RIPA may change/ quench the fluorescence and cause significant variability across replicates, and may mistakenly measure tau monomer trapped on the cell surface. Flow cytometry (after quenching/trypsinization) based on fluorescence in intact cells, or confocal microscopy, should be used to cross-validate the findings.

We have performed additional experiments with other concentrations of monomeric tau (and tau fibrils), demonstrating the robustness of the assay and the reproducibility of the results, including at the lowest concentration, as the conclusions hold under different conditions. We also showed a linear dose response, as shown above, which also indicates the robustness of the assay. In addition, we have cross-validated the data by performing additional fluorescence measurements by confocal microscopy, which are now included in revised Figures 2B & 4B, SI Figure 8 & 11. We observe some variability, as expected in any cellular assay, which is fully apparent from the figures, which show all data points and the number of replicates indicated. The conclusions are based on the statistical treatment, which take account of variability.

o Other known inhibitors of tau uptake should be used as positive controls, e.g., heparin (for HSPGs) and RAP (for LRP1), especially to help us understand the magnitude of the effects.

As suggested by this comment, we have used the two inhibitors mentioned (RAP and heparin) as controls in the tau cellular uptake assays with both monomeric and fibrillar tau. Interestingly, we showed that RAP specifically blocks the cellular uptake of monomeric tau but not heparin, while heparin specifically blocks the uptake of fibrillar tau but not RAP, suggesting two separate uptake mechanisms between the neuronal uptake of recombinant monomeric and fibrillar tau (Figure 2 and 4). These data are consistent with a recent paper showing similar results in iPSCNs neurons (see reference 13, Marvian et al, 2024). In addition, benchmarking with RAP allowed us to show that VHH H3-2 is (almost) as potent as this known inhibitor.

These additional data are included and discussed (including this very recent new reference) in the revised manuscript and figures.

o Why did the authors choose the tau isoform 1N4R over other isoforms for uptake assays? Do the VHH prevent uptake of other isoforms as well?

The tau cellular uptake assays were done with tau1N4R mainly because it is more abundant than the 2N3/4R isoform in the adult human brain (see references below). We now provide in the revised version additional assays with tau fibers, in addition to the assays with the isolated microtubule-binding domain (tauMTBD). We also provide affinity measurements for both tau1N4R and tau 2N4R, showing very similar K_D in both cases (Table1), presuming that VHHs could block the uptake of other isoforms (except A31 that would not strongly bind 3R isoforms).

References:

Hefti, M. M. *et al.* High-resolution temporal and regional mapping of MAPT expression and splicing in human brain development. *PLoS One* **13**, e0195771 (2018).

Buchholz, S. & Zempel, H. The six brain-specific TAU isoforms and their role in Alzheimer's disease and related neurodegenerative dementia syndromes. *Alzheimers Dement. J. Alzheimers Assoc.* **20**, 3606–3628 (2024).

Results

Page 6, 1st paragraph: How did the authors select for the nanobodies? We understand that this work was done in prior publications. It would be helpful to include a brief summary in the result section or method section.

A description of the initial VHH selection has been added in the method section.

Page 13: see comments on uptake assay with lysis above

Additional fluorescence measurements and visualization are provided in the revised version, using confocal microscopy, see above (page 3).

Page 15, 2nd paragraph: The authors incorrectly state that LRP1 mainly mediates tau uptake. Multiple papers in the literature show that other receptors (including HSPGs) play a significant role as well. Please also correct similar statements on page 19.

Following our new data, we have nuanced this statement, first by specifying that it concerns the soluble monomeric tau and by mentioning the important role of HSPGs:

In introduction

"Active endocytosis and micropinocytosis have been described as the main drivers of cellular uptake of tau fibrils, and this requires the involvement of heparan sulfate proteoglycans (HSPGs).^{10–13} On the other hand, active endocytosis drives the cellular uptake of monomeric tau, and the low-density lipoprotein receptor-related protein 1 (LRP1) has been proposed to be the major receptor that is responsible for its uptake.^{13,14}"

We have also included new HSPG-focused data in the revised version (heparin control, heparin inhibition of fiber uptake), and a detailed discussion of the potential predominant specific role of LRP1 for monomeric soluble tau uptake while HSPGs would be more important for the uptake of the fibers. This is indeed the observation that we have made in the experiments presented in the revised version, which are consistent with the very recent publication by Marvian et al, 2024 (reference 13).

Hopefully, all together, we are providing a more balanced view of the mediators of tau cellular uptake.

Page 16, top of page: The authors report: "VHH H3-2, which showed the strongest inhibition of cellular tau uptake, was also tested but strong unspecific binding on the chip prevented valid measurements." Could the authors comment in more detail what is meant with this statement and why there was non-specific binding?

To perform the SPR experiments, we first immobilized the Fc-Cluster III on CM5 sensor chips using NHS EDC conjugation. We then performed the competition assay by co-injecting the tau1N4R-VHH mixture to observe tau binding. The difficulty was that the VHHs alone at the highest concentration could interact non-specifically with the dextran-matrix of the sensor chip (as seen in the control channel), and especially VHH H3-2.

To overcome this technical problem of non-specific binding of the VHHs to the matrix, we performed additional experiments using a protein G sensor chip to immobilize the Fc-receptor cluster III. The control experiment to measure the kinetic parameters of tau association showed good quality data providing a K_D of 91 nM (SI Figure 13A), which is close to what has already been described in the literature (see reference 43). We also used the RAP control as competitor of tau-cluster III binding and were able to demonstrate the inhibition (SI Figure 13B). We then performed the competition assay in the presence of

the VHHs. Using this set-up, we did not detect any unspecific binding response with the injection of VHH alone.

We were able to reproduce the data that were presented in the first manuscript (with the CM5 sensor chips) and provide additional data in the revised version, clearly showing that VHH H3-2 is the strongest competitor of the tau-LRP1 (cluster III) interaction in the series of anti-tau VHHs that we have tested (Figure 5).

Page 20, middle paragraph:

- The authors demonstrate binding of tau 1N4R to cluster III domain of LRP1 *in vivo*, and provide *in vitro* data for competition of A31 and Z70mut1 with the tau:LRP1 cluster III interaction. We would suggest confirming this result *in vivo*, e.g., stain for LRP1 on the cell surface and then assess for co-localization with tau in the presence and absence of VHH.

We have not been able to image the cell surface using an antibody against LRP1 (Abcam) and some kinetics of uptake (5-15-60 min), mainly because the signal very rapidly highlights the endosomal pathway (rather than the surface).

However, we now provide additional *in vitro* data on the competition of VHH H3-2 with the tau:LRP1 clusterIII interaction.

- Given the relevance of other receptors for tau uptake, it would greatly increase the impact of this paper if the authors checked for competition of the VHH with tau:HSPG binding.

This is certainly interesting but technically difficult. Firstly, in the experiments that we have performed, we clearly conclude to the involvement of HSPGs mainly for the tau fibers, which are difficult to manipulate to measure the interaction because they are "solid". Secondly, we see only a moderate activity of VHH H3-2 alone, which may be difficult to demonstrate *in vitro*.

- Page 21, 1st and 2nd paragraph: the observation that H3-2 forms dimers that are enhanced by C-terminal tau binding is interesting. I am wondering if H3-2 somehow promotes fibrillization of tau (this is true for heparin, which is also used as a very effective uptake inhibitor of tau). The authors could test this easily by incubating H3-2 with tau monomer and checking with Thioflavin T and TEM for aggregation.

We have performed *in vitro* kinetic aggregation assays with 10 μ M of tau2N4R in the absence (Tau-, orange curve, negative control for aggregation), or in the presence of heparin (Tau+, blue curve, positive control for aggregation) monitored by ThT fluorescence, at 37°C. In the presence of 4 μ M of VHH H3-2 and without heparin (H3-2-, green curve), no aggregation is observed. We even observed a reduction in tau aggregation in the presence of heparin and of 4 μ M of VHH H3-2 (H3-2+, purple curve).

Figure 1: None of the selected VHH studies in this paper bind the N terminus, which is reportedly the binding site to the cell membrane. Could the authors comment on this?

We have no hard evidence to provide a rationale for this result, but we can confirm that all the VHHs from the tau 2N4R screen were found to bind to the C-terminal part of the protein. In the two X-ray structures of anti-tau VHHs that we were able to solve in complex with tau peptides (reference 29 + present study), the tau peptide adopts a β -strand conformation. The tendency of tau sequences to adopt a β -strand conformation is mainly found in the repeat domain (see references below 1 and 2), including what are sometimes defined as pseudo-repeats (sequences immediately adjacent to the N and C termini of the microtubule binding domain). Indeed, we found VHHs against each repeat (R1 to R4) and pseudo-repeat (R0 and R'). It is therefore possible that we have selected "conformational binders", whereas binding to unstructured peptide sequences may not be efficient enough to achieve high affinity and selection under the conditions of the screen.

Indeed, an interaction of intracellular tau with the plasma membrane via annexins has been demonstrated (see reference below 3-5). However, in this study we are investigating the interaction of extracellular tau protein (monomeric or fibrillar) with the extracellular part of membrane receptors. Finally, the cellular uptake of the six tau isoforms has already been tested in a previous study (in H4 cells, see ref 14). There was no difference in uptake efficiency between the isoforms, suggesting that the potential effect of the tau N-terminal domain on tau internalization is minimal.

References for information:

1. Mukrasch, M. D. *et al.* Structural polymorphism of 441-residue tau at single residue resolution. *PLoS Biol.* **7**, e34 (2009)
2. Mukrasch, M. D. *et al.* Sites of tau important for aggregation populate β -structure and bind to microtubules and polyanions. *J. Biol. Chem.* **280**, 24978–24986 (2005).
3. Brandt, R., Léger, J. & Lee, G. Interaction of tau with the neural plasma membrane mediated by tau's amino-terminal projection domain. *J. Cell Biol.* **131**, 1327–1340 (1995).
4. Gauthier-Kemper, A. *et al.* Annexins A2 and A6 interact with the extreme N terminus of tau and thereby contribute to tau's axonal localization. *J. Biol. Chem.* **293**, 8065–8076 (2018).
5. Brandt, R., Trushina, N. I. & Bakota, L. Much More Than a Cytoskeletal Protein: Physiological and Pathological Functions of the Non-microtubule Binding Region of Tau. *Front. Neurol.* **11**, 590059 (2020).

Figure 2B: The experimental variance for these uptake assays is quite large. This could be optimized by using flow cytometry or confocal microscopy to quantify fluorescence in intact cells. How many replicates were done per condition in each experimental run? Please include this information in the legend.

We have performed additional experiments at other concentrations of monomeric tau (and tau fibrils), demonstrating the robustness of the assay and the reproducibility of the results, including at the lowest concentration, as the conclusions stand in the diverse conditions. We also showed a linear dose response, included here above (page 2), which also indicates the robustness of the assay. In addition, we have cross-validated the data by performing additional confocal microscopy fluorescence measurements now included in revised Figures 2B & 4B, SI 8 & 11. We observe some variability, as expected in any cellular assay, which is fully appreciable from the figures were all the data points are represented and the number of replicates indicated. The conclusions are based on the statistical treatment, which take the variability into account.

Figure 2C and 3C: The quality of the confocal images is poor. It is hard to tell if labeled tau is inside of cells or not, although fluorescence is significantly reduced for A31, Z70mut1 and H3-2. Please provide higher quality images.

We provide high resolution image files, quality of the images in the manuscript pdf was indeed not optimal.

Minor comments- Introduction page 2, 1st paragraph: the majority of statements made here are missing references. Please include appropriate references.-

References have been added to the first paragraph as follows, and we have also added new references concerning the role of HSPGs in tau uptake :

“In neurodegenerative disorders, there is a hypothesis suggesting that aggregation-prone proteins may have prion-like properties.^{1,2} This hypothesis refers to the ability of certain misfolded neuronal proteins to spread/propagate their misfolding by seeding soluble homotypic protein to form protein aggregates, like observed in prion disorders.³⁻⁶”

1. Glynn, C., Rodriguez, J. A. & Hyman, B. T. The structural line between prion and 'prion-like': Insights from prion protein and tau. *Curr. Opin. Neurobiol.* **86**, 102857 (2024).
2. Zampar, S., Di Gregorio, S. E., Grimmer, G., Watts, J. C. & Ingelsson, M. 'Prion-like' seeding and propagation of oligomeric protein assemblies in neurodegenerative disorders. *Front. Neurosci.* **18**, 1436262 (2024).
3. Vazquez-Sanchez, S. *et al.* Frontotemporal dementia-like disease progression elicited by seeded aggregation and spread of FUS. *Mol. Neurodegener.* **19**, 46 (2024).
4. Pongrácová, E., Buratti, E. & Romano, M. Prion-like Spreading of Disease in TDP-43 Proteinopathies. *Brain Sci.* **14**, 1132 (2024).
5. Neupane, K. *et al.* Direct observation of prion-like propagation of protein misfolding templated by pathogenic mutants. *Nat. Chem. Biol.* **20**, 1220–1226 (2024).
6. Tullo, S. *et al.* Neuroanatomical and cognitive biomarkers of alpha-synuclein propagation in a mouse model of synucleinopathy prior to onset of motor symptoms. *J. Neurochem.* **168**, 1546–1564 (2024).

Please correct spelling/ grammar mistakes

Spelling/grammar mistakes have been corrected

Reviewer #2 (Remarks to the Author):

Reviewer #3 (Remarks to the Author):

In their work the authors investigate different single domain anti-tau antibodies or VHHs, targeting a series of different epitopes of the free soluble tau to inhibit neuronal internalization. They show which of the used VHHs is the most potent inhibitor by affinity and cellular uptake measurements. Finally, using a combination of biophysical methods, they identify VHH H3-2 as the strongest inhibitor for cellular tau uptake. The new VHHs may be useful as an immunological tool in the studies of prion-like propagation tauopathies.

The work addresses a timely and relevant problem, the manuscript is well structured and the main aims are clearly explained. The methods are clearly documented. However, further experiments are required to confirm and justify the authors' claims and conclusions before the work may be considered for publication in Nature Communications.

Specific remarks:

1.) In the NMR and affinity experiments the authors used tau2N4R, whereas in the cellular uptake and LRP1 receptor-binding experiments they instead used tau1N2R. The authors should explain why they used different tau constructs and what implications this may have had?

The tau cellular uptake assays were done with tau1N4R mainly because it is more abundant than the 2N3/4R isoform in the adult human brain (see references below). We now provide in the revised version additional assays with tau fibers, in addition to the assays with the isolated microtubule-binding domain (tauMTBD). We also provide affinity measurements for both tau1N4R and tau 2N4R, showing very similar K_D in both cases (Table1), presuming that VHHs could block the uptake of other isoforms (except A31 that would not strongly bind 3R isoforms).

References:

Hefti, M. M. *et al.* High-resolution temporal and regional mapping of MAPT expression and splicing in human brain development. *PLoS One* **13**, e0195771 (2018).

Buchholz, S. & Zempel, H. The six brain-specific TAU isoforms and their role in Alzheimer's disease and related neurodegenerative dementia syndromes. *Alzheimers Dement. J. Alzheimers Assoc.* **20**, 3606–3628 (2024).

2.) SI Fig. 2B: It looks like the concentration of the sample with tau-VHH was higher than of the reference tau. Could this affect the results of much higher intensities than 1? This does not seem to be observed in such an extent in the other spectra. Please verify.

This point is now discussed in more detail based on the basis of new data included in the revised version (SI Figure 14). Indeed, we have always observed a global reduction in tau NMR signal intensities in the presence of VHH H3-2 compared to tau alone, which is not due to a difference in tau concentration between samples. This means that it could be an effect resulting from VHH H3-2 binding to tau. It is an effect specific to VHH H3-2 binding, which is clearly demonstrated when compared to the corresponding data with VHH E2-2 (SI Figure 3), which share the same binding site.

"NMR competition assays in the presence of ^{15}N -tau, VHH H3-2 and tauMTBD, at a molar ratio of 1 : 1 : 4, showed no modification of the pattern of ^{15}N -tau resonance perturbations (SI Figure 14), confirming

the absence of a direct binding between VHH H3-2 and the MTBD, also shown by the SPR experiments (SI Figure 9). Alternatively, VHH H3-2 binding could indirectly disrupt the interaction of the LRP1 receptor with the MTBD by an effect on the conformational and/or dynamic rearrangement of the tau protein ensemble. This hypothesis may be supported by the very broad effect of VHH H3-2 binding on tau, as NMR resonance broadenings were observed for residues located further from its peptide binding site, well into the MTBD, in stark contrast to VHH E2-2, which showed only a limited set of resonance broadenings corresponding to residues located close to the binding site (SI Figures 3, 14)."

3.) SI Fig. 2A and SI Fig. 4A: In both spectra tau degradation is observed. Why did tau degradation occur and how could/does this affect the results? This is not clear.

Due to its intrinsic disorder properties, tau is sensitive to protease and degradation. Although we perform tau NMR experiments in the presence of protease inhibitors, some C-terminal degradation may occur. We have performed new NMR experiments presented in revised SI Figures 3 and 14 (replacing SI Figure 4), where no degradation is observed in the spectra. This has not affected the results of the epitope mapping, and the same conclusion regarding the VHH binding sites is presented in the revised version.

4.) line 333: SCK experiments confirmed the binding

The SCK experiments show quite different affinities of VHH A31 and Z70mut1 when using the MTBD only. The value of the K_D of A31 to tau was 116 nM compared to 41 nM to tauMTBD, but the striking difference is the affinity of Z70mut1 to tau 21 nM compared to 799 nM to tauMTBD. The authors should explain this 38-fold difference in the K_D of Z70mut1 and its influence on the binding/function.

Thank you for the comment about the outlying data.

This observed difference is not an effect due to binding to tau *versus* tauMTBD, but rather an effect due to VHH Z70mut1 *versus* VHH Z70mut1 with a c-myc C-terminal tag. As can be seen in the experiments included here, a 10-12-fold difference in affinity characterizes the binding of Z70mut1c-myc or Z70mut1 to tau2N4R, implying that the c-myc tag positively influences the affinity of the VHH Z70mut1 for tau. The data presented in the submitted manuscript compared the interaction of Z70mut1c-myc with tau2N4R (described in reference 30) with the interaction of VHH Z70mut1 with tauMTBD (this study).

Due to the influence of the tag and to compare the overall affinity parameter data, we have now performed the binding characterization by SPR of all the anti-tau VHs without any tag against tau2N4R and tau1N4R. We have added a table as experimental results in the revised version (see Table 1 and SI Figures 5 - 6 for the full SCK sensorgrams), showing no strong differences in the affinities of the VHs for the two isoforms or for the MTBD (VHs A31 and Z70mut1 to tauMTBD, with K_D s of 41 and 799 nM consistent with their respective binding affinities to tau1N4R, with K_D s of 37 and 468 nM).

In addition, the authors write that no interaction between tauMTBD and VHH H3-2 was detected, even though the NMR binding profile shows prominent signal attenuations in the repeat region as well. The authors should comment on this.

We have performed an additional NMR HSQC experiment using 100 μM of labeled ^{15}N tau, 100 μM of VHH H3-2 and 400 μM of “cold” unlabelled tauMTBD fragment. We then compared the ratio of the corresponding resonance intensities between ^{15}N tau_H3-2 (1/1 molar ratio, 100 μM each) and ^{15}N tau_H3-2_tauMTBD. Interestingly, we did not see a reduction in the broadening of the resonances of ^{15}N tau_H3-2 in the presence of the tau MTBD, compared to ^{15}N tau_H3-2, suggesting that the broadening effect observed in the MTBD region is not due to a direct interaction of VHH H3-2 with the MTBD (SI Figure 14). We hypothesize that this is due to a conformational rearrangement due to VHH H3-2 binding, as discussed in the revised version of the manuscript (see also comment above pages 8-9).

In addition, SPR experiments also did not detect any interaction between VHH H3-2 and tauMTBD, as shown in SI Fig 9 (or the isolated PHF6 or PHF6* peptides, see SPR experiments included here page13 for reviewers view only).

5.) In the binding experiment to cluster III of the LRP1, VHH A31 and Z70mut1 together with tau were measured but not H3-2, which showed the strongest inhibition of cellular uptake. What could be the cause of the unspecific binding of the VHH H3-2 to the chip? Could it be the formation of a dimer? Is it possible to try different concentrations or conditions in order to measure/detect H3-2 as well? In my opinion this is an important experiment that would provide evidence that may (or may not) justify the conclusions for H3-2 as well.

To perform the SPR experiments, we first immobilized the Fc-Cluster III on CM5 sensor chips using NHS EDC conjugation. We then performed the competition assay by co-injecting the tau1N4R-VHH mixture to observe tau binding. The difficulty was that the VHHs alone at the highest concentration could interact non-specifically with the dextran-matrix of the sensor chip (as seen in the control channel), and especially VHH H3-2.

To overcome this technical problem of non-specific binding of the VHHs to the matrix, we performed additional experiments using a protein G sensor chip to immobilize the Fc-receptor cluster III. The control experiment to measure the kinetic parameters of tau association showed good quality data providing a K_D of 91 nM (SI Figure 13A), which is close to what has already been described in the literature (see reference 43). We also used the RAP control as competitor of tau-cluster III binding and were able to demonstrate the inhibition (SI Figure 13B). We then performed the competition assay in the presence of the VHHs. Using this set-up, we did not detect any unspecific binding response with the injection of VHH alone.

We were able to reproduce the data that were presented in the first manuscript (with the CM5 sensor chips) and provide additional data in the revised version, clearly showing that VHH H3-2 is the strongest competitor of the tau-LRP1 (cluster III) interaction in the series of anti-tau VHHs that we have tested (Figure 5).

6.) line 506: Because of this specific binding mode, VHH H3-2 could indirectly disrupt the interaction of a LRP1 receptor with tauMTBD, ... Please provide additional experiments supporting this claim, so far it seems speculative.

- We have added new data to confirm that there is no direct binding of VHH H3-2 to the MTBD domain (SPR data and NMR data included in the revised manuscript) or to the isolated PHF6 or PHF6* (see data included below page 13). We have added new SPR data showing that *in vitro* VHH H3-2 can compete with tau2N4R binding to Cluster III-LRP1 binding.

- We show the direct uptake of tauMTBD and the ability of both VHH A31 and Z70Mut1 to decrease the uptake and to compete with tau2N4R binding to Cluster III-LRP1 binding. According to the literature K_D s of tau2N4R-LRP1 and tauMTBD-LRP1 are very close (K_D of 60 ± 8 nM and 73 ± 18 nM, respectively), suggesting that tauMTBD is sufficient for tau-LRP1 binding (see reference 43). We found a very similar K_D of $91 \text{ nM} \pm 1 \text{ nM}$, SI Figure 13, for tau2N4R.

- We show comparison of VHH E2-2 and VHH H3-2 that share the same epitope, and yet have very different abilities to inhibit tau uptake. The NMR spectra due to the binding of each of these VHHs is strikingly different: in the case of VHH E2-2 binding, the signal broadening is restricted to resonances of residues close to the binding site (SI Figure 3).
- In addition, despite the fact that VHH H3-2 does not bind within the MTBD, we observe signal broadening in the NMR spectra of ¹⁵N-tau, corresponding to resonances of residues located in regions within the MTBD (SI Figure 14).

We propose this hypothesis only in the discussion section, based on all these data, including new experimental data added in the revised version to strengthen this hypothesis.

Minor remarks

The following remarks have been taken into account

- 1.) line 307: tau uptake
- 2.) line 628: Please correct the title
- 3.) line 3 in SI: Please correct to Figure 1 (now Fig. 2)
- 4.) It would be informative to have the sequence alignment of all three C-ter binding VHH (in the SI Fig. The revised figure now shows a sequence alignment of the three VHHs (SI Figure 4)
- 5.) Please arrange the citation of the SI Figures in the manuscript in the order they appear in the Supplementary Material.
- 6.) SI Fig. 6. Please correct the title

Reviewer #4 (Remarks to the Author):

A very interesting study on inhibition of neuronal internalization of free soluble tau by VHH antibodies. While several VHH were found to bind to various epitopes in tau2N4R, VHH A31, Z70mut1 and H3-2 were found to strongly inhibit tau cellular internalization. Two of these also competed with the LRP1 receptor. Three VHH, F8-2, E2-2, and H3-2, were found to bind to the C-terminal domain with F8-2 binding weaker in the μM range and H3-2 more strongly in the low nM range. H3-2 was the most potent in inhibiting cellular tau uptake, with E2-2 being intermediate, whereas F8-2 was inactive. The inhibition was somewhat correlated with their relative epitope binding affinities. A crystal structure was obtained for VHH H3-2 with the tau C-terminal peptide (residues 369-381) and the VHH was found to unexpectedly form a domain-swapped dimer where the CDR H3 and FR4 were swapped with their counterparts in the VHH dimer and the CDR H3 formed an anti-parallel β -sheet. To rule out crystallization artifacts, the H3-2 VHH was analyzed in solution by size exclusion chromatography and was found as a predominantly dimeric form in the presence and absence of tau peptide compared to E2-2, which formed a monomer. Thus, there is supporting evidence for F8-2 forming a biological dimer. I will focus mainly on the structural work and have the following comments.

Page 10, line 229. Please subscript all of the 1's in P212121 to read P₂₁₂₁₂₁.

This was edited in the revised version.

Page 12, line 277. Please replace "complementary" with "complementarity".

This was replaced in the revised version.

Page 18, lines 417-418. "The interaction is mainly driven by main-chain hydrogen bonds between the peptide and the CDR3 in a parallel β -sheet conformation, forming a 4-stranded β -sheet with 2 CDR3 and 2 peptides". So where does the sequence specificity for this particular peptide arise if the main interactions are to the peptide backbone? There needs to be more discussion concerning this point.

Do other peptides bind to this domain-swapped dimer?

Indeed, the structure of the complex and the high-resolution interaction do not give any obvious indication of the sequence specificity for the tau C-terminal peptide. However, we can provide further evidence of specificity. We therefore performed a concentration response of different tau peptides from different tau regions on VHH H3-2 immobilized on a SA sensor CHIP as described in the manuscript and using the same tau peptide concentration. R0 corresponds to peptide REPKKVAVVRTP localized in the proline-rich domain of tau. PHF6* corresponds to peptide PGGGKVQIINKKLDLSNK localized in the R2 tau region. PHF6 corresponds to the peptide PGGGSVQIVYKPVDSLK located in the R3 tau region. Only the tau-C-ter peptide we described in the present study (KKIETHKLTREN) interacts with VHH H3-2, reinforcing our conclusion about the specificity of VHH H3-2 for this particular tau C-ter epitope. Of note, the published crystal structure of another anti-tau VHH (VHH Z70) in complex with the PHF6 sequence shows a similar intermolecular interaction limited to the backbone, as well as the formation of a β -sheet between the PHF6 peptide sequence and the CDR3 (but not a swapped dimer, ref 29). Nevertheless, this VHH is so specific that it has a low affinity for the very similar peptide sequence PHF6*. In both cases we can assume that the VHHs recognize a specific conformation that requires a specific complementarity to form a β -sheet. We recognize that this remains speculative.

Does peptide also bind to the monomeric VHH?

We have no direct evidence for a positive interaction of tau with a monomeric VHH H3-2, and there is no easy way to investigate this point because, as we have shown using the SEC experiment, VHH H3-2 can already adopt a dimeric conformation in solution without the presence of the peptide. When we immobilized VHH H3-2 on the sensor CHIP or on a gold particle, we still cannot exclude the formation of the dimer.

Pages 21-22, lines 495-496. "The only difference between VHHS F8-2 and E2-2 are their CDR1 and CDR2 sequences, which only modulate their affinity for tau, not their epitope". Sorry I don't follow this. Is the epitope not part of tau? What are the CDR1 and CDR2 sequences recognizing - or does the CDR1 and 2 sequences only indirectly influence binding and epitope affinity.

Indeed, it is known that the CDR3 loop mainly drive the epitope recognition, meaning that VHHS sharing the same CDR3 loop recognize identical epitope. From our previous work (ref 28), including this study, we have shown that CDR1 and 2 can modulate affinity since E2-2 shows a 10 times better affinity than F8-2.

We have modified the text in the introduction and discussion parts to be clearer with appropriate reference (Ref 42), as follow:

"VHH E2-2 binds the same C-terminal epitope than VHH F8-2, as they share the same complementarity determining region 3 (CDR3) sequence (SI Figure 3-4).^{28,42} "

"In parallel, VHHS F8-2 and E2-2 which also target the tau C-terminal domain do not block tau cellular uptake. VHH H3-2 shares the same epitope, although its sequence is completely different (SI Figure 4),..."
 42. De Genst, E. *et al.* Strong in vivo maturation compensates for structurally restricted H3 loops in antibody repertoires. *J. Biol. Chem.* **280**, 14114–14121 (2005).

Page 21, lines 499-500. "differences observed for the VHHS E2-2 and H3-2, because their affinities are within the same range, as shown by SPR" I don't know that I would agree with that. There is still a four to five fold difference in Kd, and 4 fold difference in the kon.

Thank you for this comment. We have indeed changed the related discussion, according to new SPR KDs and competition data added in the revised version, which show competition with VHH H3-2, but not VHH E2-2, for tau binding to LRP1Cluster III :

“VHH H3-2 shares the same epitope, although its sequence is completely different (SI Figure 3), but its affinity for tau is 9 and 67-fold higher than that of VHHs E2-2 and F8-2, respectively (Table 1). These data therefore suggest that affinity is an important contributor to the activity of the VHHs in blocking cellular uptake of tau, but other factors may also contribute. Indeed, VHH H3-2 showed the strongest competitive effect on tau-LRP1 binding *in vitro* in SPR assays, despite the fact that the C-terminal domain of tau is not described to interact with LRP1, whereas VHH E2-2 remained inefficient.”

What is the binding affinity of TauMTBD to receptor and are these different VHH biological activities related to that?

According to the literature, K_D s of tau2N4R-LRP1 and tauMTBD-LRP1 are very close (K_D of 60 ± 8 nM and 73 ± 18 nM, respectively; see ref 43). This is now discussed in the revised version of the manuscript:

“ The membrane receptor LRP1, has been shown to interact directly, and with similar affinities, with both full-length tau and the tauMTBD fragment, suggesting that the tauMTBD is sufficient to mediate tau binding to LRP1.^{14,43} We confirmed these results using the cluster III domain of LRP1, with the same range of reported affinities under the conditions of our assay (SI Figure 13). The *in vitro* competition of VHHs A31 and Z70mut1 with tau for the interaction with LRP1 cluster III (Figure 5) suggested that they could act as tau-LRP1 competitors at the cell membrane and prevent tau receptor-mediated endocytosis. The strongest effect was observed with VHH A31, which could be explained by the 10-fold strongest affinity for tau1N4R compared to VHH Z70mut1 (Table 1).”

Page 32. There is no crystallographic table that I could find and, hence, the paper should not be accepted without it – sorry if I missed it somehow but I also could not also see any reference to tables in the text. This table needs to be reviewed. There was however a PDB report but the Ramachandran outliers were in the red region (unfavorable) as well as the RSRZ outliers. I would expect that a 1.8 Å structure would not have such a poor Ramachandran outlier score without some explanation. The data and refinement statistics in the PDB stats are also incomplete. There are no values for Rmerge in the outer shell, no Rpim, no B-factors for tau peptide compared to the VHH or overall. These should be included in a data processing and refinement statistics SI table.

The full table of statistics is now available as SI Table 1. The Ramachandran outliers are all contained within poorly defined loop regions including CDR1 and 2. To keep the main chain as complete as possible, these regions were still modeled but resulted in poor Ramachandran statistics.

“While the interaction is well defined in the structure, some loop regions, including CDR1 and 2, (26-35, 54-60, 76-81, 114-121) were poorly resolved and contains Ramachandran outliers.”

Page 34, Reference 1. Incomplete reference. Page 35, Reference 21. Incomplete reference. Page 37, Reference 41. Incomplete reference. Page 37, Reference 46. Incomplete reference. Page 38, Reference 53. Incomplete reference. Page 38, Reference 54. Incomplete reference.

All incomplete references have been corrected

**Please find below our point by point rebuttal to the last reviewer's comments.
The reviewers comments are in black, our replies in blues.**

REVIEWERS' COMMENTS

Reviewer #1 (Remarks to the Author): I am satisfied that the authors have addressed reviewer concerns sufficiently for the manuscript to be published.

Reviewer #2 (Remarks to the Author): I co-reviewed this manuscript with one of the reviewers who provided the listed reports. This is part of the Nature Communications initiative to facilitate training in peer review and to provide appropriate recognition for Early Career Researchers who co-review manuscripts.

Reviewer #3 (Remarks to the Author):
The authors have addressed my questions.

Reviewer #4 (Remarks to the Author):
Almost all of my previous comments have been satisfactorily addressed in the revision. However, I have some other minor comments that stem from the revisions. It would have also been nice to have the page numbers and lines for the revisions to the text so not to have to search through the paper looking for them as the search function doesn't seem to work so well on PDF files.

Page 14. "The full table of statistics is now available as SI Table 1.

I have some **comments on the SI Table 1**

- Resolution range. 38.17 - 1.8 (1.864 - 1.8)
- Please replace first "1.8" with "1.80" and second with "1.800" if that is what it is so the number of decimal points are the same in each category? "
- Unit cell. "70.394 76.346 95.565" Two decimal points are sufficient.
- Mean I/sigma(I). "26.27 (3.40)". One decimal point is sufficient.
- Wilson B-factor R-merge. Please add units after B-factor (Å) and change the value to an integer - decimal points are not meaningful.
- R-merge, R-meas, Rpim. 3 decimal points are sufficient.
- R-work, R-free. 3 decimal points are sufficient.
- Please replace "RMS(bonds), RMS(angles)" with "RMS Bond lengths (Å), RMS Bond angles (°)".
- Please replace "Average B-factor" with "Average B-factor (Å²)".
- macromolecules, solvent, VHHs, peptides. Please truncate all B values to integers.

Page 14, "All incomplete **references** have been corrected".

However, not all are complete, see below.

Page 32, line 776, Reference 30. Please replace "300, (2024)" with "300, 107163 (2024)".

Page 32, line 778, Reference 31. Please replace "eLife 5, (2016)" with "eLife, 5, e16228 (2016)".

Page 34, line 834, Reference 57. Please replace "7," with "7, e36584".

Final modifications have been made according to reviewer's 4 last minor comments (Supplementary Table S1, and references 30, 31 and 57).